# Interfacing nickel nitride and nickel boosts both electrocatalytic hydrogen evolution and oxidation reactions

Fuzhan Song[1], Wei Li [1], Jiaqi Yang[2], Guanqun Han[1], Peilin Liao[2] & Yujie Sun [1,3]

Electrocatalysts of the hydrogen evolution and oxidation reactions (HER and HOR) are of critical importance for the realization of future hydrogen economy. In order to make electrocatalysts economically competitive for large-scale applications, increasing attention has been devoted to developing noble metal-free HER and HOR electrocatalysts especially for alkaline electrolytes due to the promise of emerging hydroxide exchange membrane fuel cells. Herein, we report that interface engineering of $Ni_3N$ and Ni results in a unique $Ni_3N$/Ni electrocatalyst which exhibits exceptional HER/HOR activities in aqueous electrolytes. A systematic electrochemical study was carried out to investigate the superior hydrogen electrochemistry catalyzed by $Ni_3N$/Ni, including nearly zero overpotential of catalytic onset, robust long-term durability, unity Faradaic efficiency, and excellent CO tolerance. Density functional theory computations were performed to aid the understanding of the electrochemical results and suggested that the real active sites are located at the interface between $Ni_3N$ and Ni.

[1] Department of Chemistry & Biochemistry, Utah State University, Logan, UT 84322, USA. [2] School of Materials Engineering, Purdue University, West Lafayette, IN 47907, USA. [3] Department of Chemistry, University of Cincinnati, Cincinnati, OH 45221, USA. These authors contributed equally: Fuzhan Song and Wei Li. Correspondence and requests for materials should be addressed to P.L. (email: lpl@purdue.edu) or to Y.S. (email: yujie.sun@uc.edu)

Hydrogen ($H_2$) has long been advocated as a clean and carbon-neutral energy carrier in the field of renewable energy catalysis, in that $H_2$ can be produced from water electrolysis with renewable energy inputs, like solar and wind power, and its utilization in hydrogen fuel cells will produce electricity with water as the sole product[1]. The success of a future hydrogen economy strongly depends on the efficient $H_2$ production and utilization, which includes the hydrogen evolution and oxidation reactions (HER and HOR)[2–5]. Owing to the multi-proton multi-electron nature of both HER and HOR, electrocatalysts are indispensable to drive the two reactions to achieve industrially relevant rates. Pt-based electrocatalysts exhibit the best performance for $H_2$ evolution in strongly acidic electrolytes[6], however their HER activities are substantially diminished under alkaline conditions. Since no Earth-abundant electrocatalysts of water oxidation can survive under strongly acidic conditions and match the rates of Pt-based HER electrocatalysts so far, an increasing attention has been shifted towards $H_2$ evolution in alkaline media, in which a number of low-cost HER electrocatalysts start to rival Pt-based HER electrocatalysts. The same scenario occurs for the $H_2$ oxidation reaction. Pt is still the state-of-the-art HOR electrocatalyst under acidic conditions for the application of proton exchange membrane fuel cells (PEMFCs)[7–9]. However, the real kinetic bottleneck of PEMFCs in acidic electrolytes is the cathodic $O_2$ reduction reaction (ORR), which requires a large amount of unaffordable Pt. In order to develop economically attractive hydrogen fuel cells, it is imperative to develop competent fuel cell electrocatalysts composed of much fewer or no Pt-group metals. Recently, hydroxide exchange membrane fuel cells (HEMFCs) emerge as a promising alternative technology[5,7–12], whose alkaline electrolytes enable the utilization of many inexpensive ORR electrocatalysts, some of which can compete the performance of Pt-based ORR electrocatalysts. Ironically, under alkaline condition, it is HOR, instead of ORR, becoming the challenging reaction, as even for Pt its HOR performance in alkaline HEMFCs is two orders of magnitude lower than that in acidic PEMFCs. Therefore, it is of fundamental and practical importance to develop highly competent and Earth-abundant electrocatalysts for improving hydrogen electrochemistry in both HER and HOR for the realization of hydrogen economy[2,13].

Great research efforts have been devoted to the development of nonprecious HER electrocatalysts, including transition metal compounds, alloys, and molecular complexes[14–19]. Relatively less attention has been concentrated on the development of HOR electrocatalysts[8,9,20–23]. Since both HER and HOR involve the same critical intermediate species, adsorbed hydrogen ($H^*$) on the surface of an electrocatalyst, it is not surprising that hydrogen adsorption free energy ($\Delta G_{H^*}$) has been widely adopted as a key descriptor in assessing the performance of diverse electrocatalyst candidates for HER and HOR[6,24,25]. The accumulated collection of experimental and theoretical results has unambiguously established volcano-type plots for HER/HOR activity versus $\Delta G_{H^*}$ on many electrocatalysts, indicating that the optimal HER/HOR performance will be achieved when $\Delta G_{H^*}$ is near 0 eV[3,26,27]. Hence, great efforts have been focused on optimizing $\Delta G_{H^*}$ of diverse electrocatalysts through metal alloying[9,15,25], composition variation[21,22,28], crystal facet modification[17], defect introduction, size/dimension confinement[29], and interface construction[7,8,20,30–36]. Despite the increasing efforts in advancing the HER and HOR activities of inexpensive electrocatalysts, most of them have not met the target performance for large-scale industrial applications. To the best of our knowledge, no catalytic systems ever reported focus on exploring the interfaces of first-row transition metals and their nitrides for hydrogen electrochemistry in aqueous media.

Herein, we demonstrate that purposely interfacing Ni and $Ni_3N$ results in an electrocatalyst ($Ni_3N/Ni$) with extraordinary activities for both HER and HOR. The rich $Ni_3N/Ni$ interfacial sites can be obtained by electrodeposition of Ni nanoparticles on current collectors such as Ni foam (NF) followed by thermal nitridation in ammonia ($Ni_3N/Ni/NF$). Through interface engineering, the resultant $Ni_3N/Ni/NF$ demonstrates excellent HER apparent activity with nearly zero onset overpotential in alkaline and neutral electrolytes, requiring only 12 to 19 mV overpotential to produce a current density of $-10\,mA\,cm^{-2}$, which can rival the activity of Pt/C catalyst loaded on NF under the present experimental conditions. Such exceptional electrocatalytic performance renders $Ni_3N/Ni/NF$ the best among all the reported nonprecious HER electrocatalysts. Besides, the intrinsic specific activities (normalized by the real surface area or electrochemically active surface area) of $Ni_3N/Ni/NF$ are also superior to those of Pt/C catalysts loaded on NF for HER in neutral and alkaline electrolytes under similar experimental conditions within the scope of our investigation. Even more exciting is that $Ni_3N/Ni/NF$ also shows superior HOR activity in alkaline medium (0.1 M KOH) with a great tolerance to CO poisoning. Density functional theory calculations were conducted to shed light on the exceptional performance of $Ni_3N/Ni/NF$. It was found that the interfacial sites between $Ni_3N$ and Ni have very small values of $\Delta G_{H^*}$. The best hydrogen adsorption site on $Ni_3N/Ni/NF$ exhibits a $\Delta G_{H^*}$ value of 0.01 eV, very close to the ideal amount of 0 eV. Furthermore, our computational results also imply that the existence of a $Ni_3N/Ni$ interface favors both the original adsorption and the subsequent dissociation of water on the catalyst surface, which is beneficial to HER (and arguably HOR as well) activity in alkaline and neutral electrolytes. Overall, $Ni_3N/Ni/NF$ represents an extremely active while still low-cost electrocatalyst with bifunctional activity for both HER and HOR. Our work also demonstrates that interfacing metals and nitrides is an effective strategy in creating inexpensive and high-performance catalysts of hydrogen electrochemistry, which deserves further attention for applications not only limited to water electrolyzers and fuel cells but also many other hydrogen-related reactions.

## Results

**Synthesis and characterization of interfacial $Ni_3N/Ni$.** The $Ni_3N/Ni$ interfacial electrocatalysts were synthesized through the cathodic electrodeposition of porous Ni microspheres on common current collectors such as Ni foam (NF) or carbon foam (CF), followed by thermal nitridation in ammonia to create rich $Ni_3N/Ni$ interfacial sites. The nitridation temperature and duration were both optimized for $Ni_3N/Ni/NF$. The nitridation temperature was varied from 200 to 400 °C and no $Ni_3N$ was formed until the temperature reached 300 °C (Supplementary Fig. 1). Further increasing the nitridation temperature led to the disappearance of the $Ni_3N$ phase, most likely due to the low thermal stability of $Ni_3N$ at high temperature (>350 °C)[37,38]. The nitridation duration was also varied at 300 °C (Supplementary Fig. 2). On the basis of the weight increase after ammonia treatment, the weight percentage of $Ni_3N$ in $Ni_3N/Ni/NF$ increased from 8.67 to 44.66 wt.% with the nitridation duration rising from 0.5 to 12 h, indicating the increased coverage of $Ni_3N$ (Supplementary Fig. 3). The $Ni_3N/Ni/NF$ synthesized at 300 °C for 6 h with the weight percentage of $Ni_3N$ of 41.82 wt.% exhibits the highest HER activity, while longer duration resulted in decreased HER activity (see Discussion).

The scanning electron microscopy (SEM) images in Fig. 1a illustrate that $Ni_3N/Ni/NF$ prepared at 300 °C for 6 h possessed three-dimensional (3D) macroporous ligament network structure

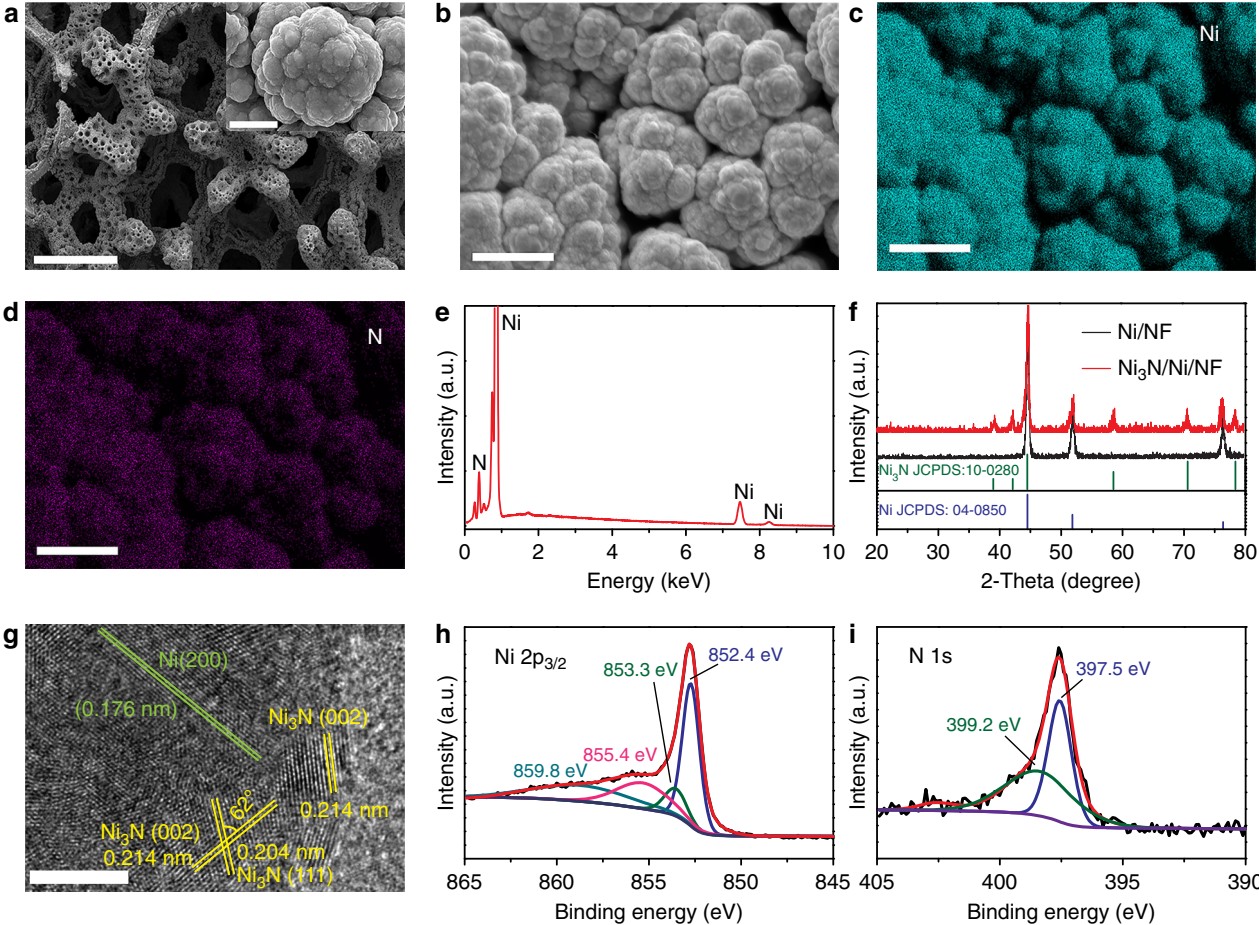

**Fig. 1** Characterization of Ni₃N/Ni interfacial electrocatalysts. **a**, **b** SEM images of Ni₃N/Ni/NF at different magnifications. Elemental mapping images of Ni (**c**) and N (**d**). Scale bars, 500 μm (**a**); 3 μm inset of **a**; 10 μm (**b**–**d**). **e** EDX spectrum of Ni₃N/Ni/NF. **f** XRD patterns of Ni₃N/Ni/NF and Ni/NF. **g** HRTEM image of Ni₃N/Ni interface. Scale bar, 5 nm (**g**). **h**, **i** XPS spectra of Ni 2p$_{3/2}$ (**h**) and N 1s (**i**)

with numerous stacked coarse particles over the skeleton surface, which is inherited from the electrodeposited Ni/NF sample yet in sharp contrast to the smooth surface of pristine Ni foams (Supplementary Fig. 4). The elemental mapping images of Ni₃N/Ni/NF show that Ni and N are uniformly distributed and the energy dispersive X-ray (EDX) spectrum confirms the major composition of Ni and N in Ni₃N/Ni/NF (Fig. 1b–e). The X-ray diffraction (XRD) patterns (Fig. 1f) suggest that after nitridation new peaks attributed to hexagonal Ni₃N (JCPDS card No. 10-0280) appeared while the major composition of Ni₃Ni/Ni/NF remained as the cubic Ni phase (JCPDS card No. 04-0850)[38]. The high-resolution transmission electron microscopy (HRTEM) image of the Ni₃N/Ni interfacial electrocatalyst clearly shows the interface between hexagonal Ni₃N and cubic Ni (Fig. 1g). The well-resolved lattice fringes with inter-planar spacing of 0.204 and 0.214 nm can be unambiguously assigned to the (111) and (002) crystal planes of hexagonal Ni₃N with an intersection angle of 62°[39,40], in agreement with the XRD results. The unique lattice fringes with inter-planar distance of 0.176 nm correspond to the (200) crystal plane of cubic Ni. Moreover, the elemental mapping results of N and Ni (Supplementary Fig. 5) demonstrate that Ni is homogeneously distributed, while N is sporadically located. The HRTEM and elemental mapping results corroborate the successful formation of rich Ni₃N/Ni interfaces. The surface elements and their valence states in Ni₃N/Ni/NF were further probed by X-ray photoelectron spectroscopy (XPS). As shown in Fig. 1h, the high-resolution Ni 2p$_{3/2}$ spectrum can be deconvoluted to features with maxima at 852.4 and 853.3 eV, which are assignable

to metallic Ni and Ni(I) of Ni₃N, respectively[41–44]. A small peak at 855.4 eV corresponds to the oxidized Ni species likely due to adventitious surface oxidation; while the satellite peak at 859.8 eV is attributed to the shake-up excitation of the high-spin nickel ions[42]. The N 1s XPS spectrum in Fig. 1i can be simulated by the combination of two features at 397.5 and 399.2 eV, ascribed to N species of Ni₃N and NH moieties, respectively, in which the latter likely resulted from the incomplete reaction with NH₃[42,45,46]. Due to the partial transformation of surface Ni to Ni₃N, the collective characterization results discussed above suggest that Ni₃N/Ni/NF inevitably possesses rich interfacial sites between Ni and Ni₃N.

**Electrocatalytic H₂ evolution.** The electrocatalytic performance of Ni₃N/Ni/NF towards HER was investigated in H₂-saturated electrolytes of 1.0 M potassium phosphate (KPi) buffer (pH 7.17) and 1.0 M KOH (pH 13.80). All potentials reported herein are referenced to the reversible hydrogen electrode (RHE) and the current densities were calculated on the basis of both the geometric areas and the real surface areas of electrodes. The effects of nitridation temperature and duration of Ni₃N/Ni/NF on the electrocatalytic HER activity were first studied. As shown in Supplementary Figs. 6–7, the Ni₃N/Ni/NF synthesized at 300 °C for 6 h possessed the best HER activity in all electrolytes, highlighting the importance in obtaining the appropriate amount of Ni₃N/Ni interfacial sites for optimal HER performance. Therefore, all the following studies were conducted on Ni₃N/Ni/NF prepared at 300 °C for 6 h unless noted otherwise.

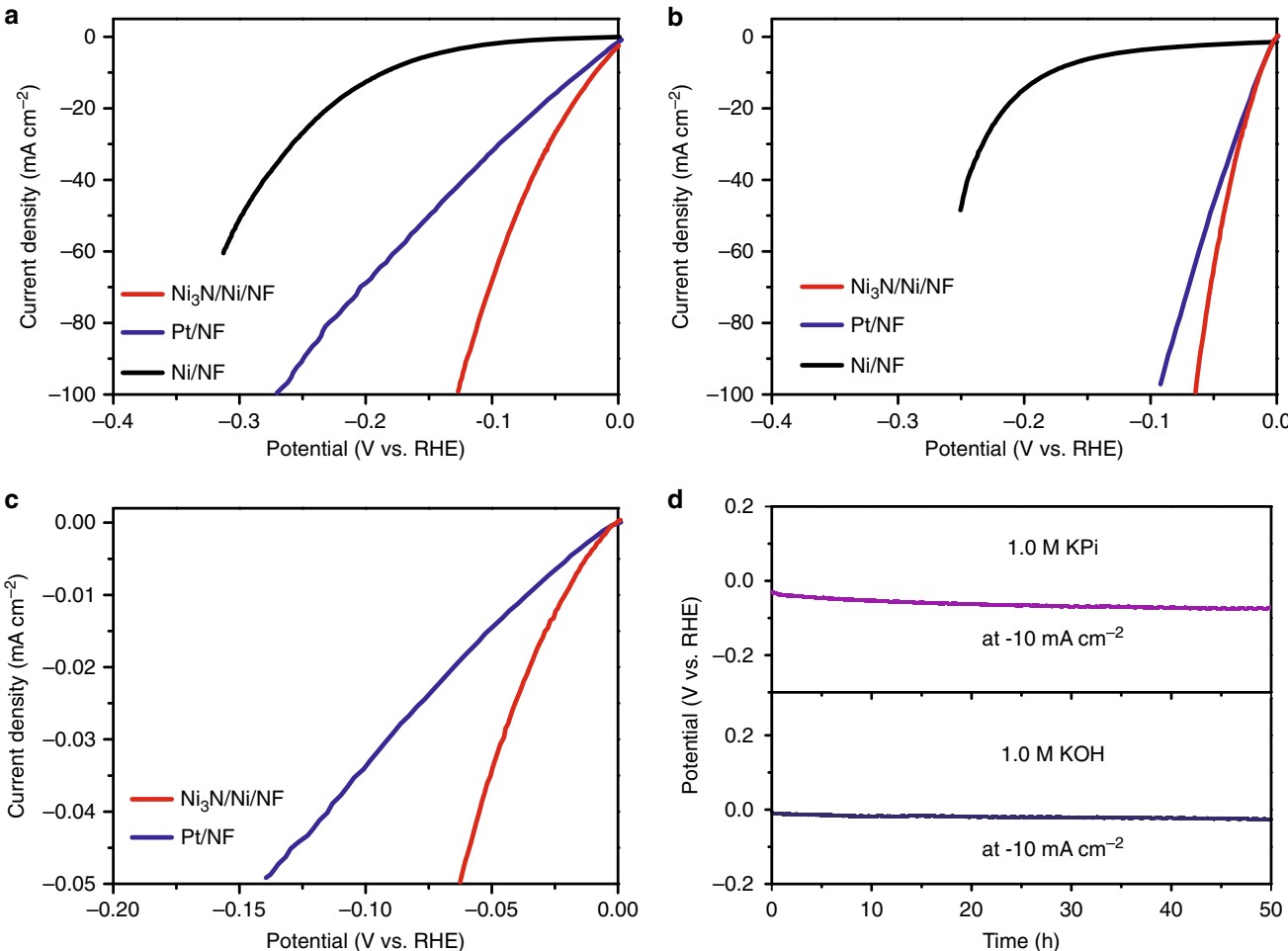

**Fig. 2** The electrocatalytic HER performance in neutral and alkaline solutions. **a**, **b** Linear sweep voltammetry (LSV) curves of Ni₃N/Ni/NF, Ni/NF, and optimized Pt/NF (Pt/C: 2.5 mg cm⁻²) for HER in 1.0 M potassium phosphate (KPi) buffer (**a**) and 1.0 M KOH (**b**) with the current density normalized by the geometric electrode area. **c** LSV curves of Ni₃N/Ni/NF and Pt/NF for HER in 1.0 M KOH with the current density normalized by their respective BET surface areas. All LSV curves are iR-corrected. **d** Chronopotentiometry curves of Ni₃N/Ni/NF collected at −10 mA cm⁻² normalized by the geometric electrode area in 1.0 M KPi (top) and 1.0 M KOH (bottom) without iR correction

For comparison, Pt/C (20 wt.%) powder loaded on the same-type nickel foam was included as a control sample (Pt/NF). The loading amount of Pt/C (2.5 mg cm⁻²) was optimized to achieve the best electrocatalytic HER activity under each pH condition (Supplementary Figs. 8–9). The iR-corrected linear sweep voltammetry (LSV) curves of Ni₃N/Ni/NF, Ni/NF, and Pt/NF for H₂ evolution at different pH are plotted in Fig. 2a, b. Under neutral and alkaline conditions, because of the lack of free protons, water adsorption and dissociation generally take place prior to H₂ evolution[2,5,30,47]. To our delight, Ni₃N/Ni/NF exhibited extraordinary HER performance with catalytic onset potentials at ~0 V vs. RHE and required an overpotential of only 19 and 12 mV to deliver a current density of −10 mA cm⁻² in 1.0 M KPi buffer and 1.0 M KOH, respectively. In 1.0 M KPi, in order to produce an industrially meaningful current density like −100 mA cm⁻², Ni₃N/Ni/NF only needed an overpotential of 126 mV. However, an overpotential of 272 mV was required for Pt/NF to deliver the same HER current. Without nitridation, the parent Ni/NF showed rather mediocre HER activity. The drastic difference in their HER activities of Ni₃N/Ni/NF and Ni/NF unequivocally proves the critical role of the Ni₃N/Ni interfaces formed during nitridation in catalyzing H₂ production, as these two electrodes have similar morphology. The best HER performance of Ni₃N/Ni/NF was achieved in 1.0 M KOH, wherein an overpotential of

merely 64 mV was needed to produce −100 mA cm⁻², saving at least 31 mV of voltage input relative to that on Pt/NF. The charge transfer resistance of Ni₃N/Ni/NF was also much smaller than that of Ni/NF under the same conditions (Supplementary Figs. 10–11).

In order to assess the intrinsic specific activities of Ni₃N/Ni/NF and Pt/NF, their electrocatalytic activities were normalized by the Brunauer–Emmett–Teller (BET) surface area measured by N₂ adsorption–desorption[48] and the electrochemically active surface area (ECSA) measured by the double layer capacitance method on the basis of cyclic voltammetry in a nonaqueous aprotic KPF₆–CH₃CN electrolyte (Fig. 2c, Supplementary Figs. 12–16 and Tables 1-2)[49]. Apparently, Ni₃N/Ni/NF showed higher specific activity than Pt/NF for HER in alkaline and neutral electrolytes under similar experimental measurement conditions. The Tafel plots of Ni₃N/Ni/NF, Pt/NF, and Ni/NF derived from their respective polarization curves in 1.0 M KOH presented overpotential-dependent Tafel slopes (Supplementary Fig. 17). The findings of variable Tafel slopes were reported for many HER electrocatalysts including Co₀.₆Mo₁.₄N₂[45], Pt[50–52], Ni-Mo-Cd[53], Ni₂P[54,55], MoP[56], and FeP/Ni₂P[57] under different pH conditions. This phenomenon could be attributed to many factors, such as back reaction at low overpotentials, mass transport together with the blocking effect of produced H₂ bubbles at high overpotentials,

formation of a large number of N–H moieties, and the dependence of adsorbed hydrogen intermediate on overpotential[45,50,58]. Therefore, it is difficult to ascertain the rate determining step(s) and kinetic mechanism of $Ni_3N/Ni/NF$ for HER from its potential-dependent Tafel slopes. Future work will aim to elucidate the catalytic mechanism of $Ni_3N/Ni$ with more sophisticated electrochemical techniques[47,59].

The exciting HER activity of $Ni_3N/Ni/NF$ prompted us to further evaluate its durability for long-term $H_2$ production through repetitive cyclic voltammetry (CV) and chronopotentiometry experiments. After 5000 CV cycles in 1.0 M KPi and 10,000 CV cycles in 1.0 M KOH, $Ni_3N/Ni/NF$ only showed a slight overpotential increase by ca. 9 mV for delivering $-100\ mA\ cm^{-2}$ under both neutral and alkaline conditions (Supplementary Figs. 18–19). As plotted in Fig. 2d, $Ni_3N/Ni/NF$ also demonstrated very stable potential requirement over 50 h of galvanostatic electrolysis at $-10\ mA\ cm^{-2}$ in both neutral and alkaline electrolytes. It could also produce a high current density of $-100\ mA\ cm^{-2}$ over 10 h with negligible degradation in 1.0 M KOH (Supplementary Fig. 20). Post-electrolysis characterization confirmed that $Ni_3N/Ni/NF$ retained its original morphology and exhibited negligible changes of morphology, crystallinity, and composition after extended HER electrolysis (Supplementary Figs. 21–23), highlighting its outstanding structural robustness and mechanical stability. The produced $H_2$ amount well matched the theoretically calculated quantity (Supplementary Fig. 24) assuming that all the passed charge was utilized to generate $H_2$, implying a Faradaic efficiency close to 100%.

In order to reveal the roles of Ni foam and Ni/NF, thermal nitridation was also conducted on either carbon foam (CF) with pre-electrodeposited Ni microparticles or bare Ni foam to obtain two control samples of $Ni_3N/Ni/CF$ and $Ni_3N/NF$, respectively. The comprehensive characterization of $Ni_3N/Ni/CF$ confirmed the presence of $Ni_3N/Ni$ interfaces over CF with the composition, morphology, and crystallinity similar to those of $Ni_3N/Ni/NF$ (Supplementary Figs. 25–27). Electrochemical studies revealed that $Ni_3N/Ni/CF$ exhibited nearly identical HER activity as $Ni_3N/Ni/NF$ (Supplementary Figs. 28–29), demonstrating that the $Ni_3N/Ni$ interfacial sites are the real active sites of HER and their activities are independent of the electrode support. In sharp contrast, $Ni_3N/NF$ has smooth surface (Supplementary Fig. 30) and shows much lower electrocatalytic HER activities in alkaline and neutral solutions (Supplementary Figs. 31–32), indicating the advantages of using rough and porous Ni/NF for thermal nitridation to obtain $Ni_3N/Ni/NF$.

Overall, the low cost, exceptional activity, and robust durability of $Ni_3N/Ni/NF$ (and $Ni_3N/Ni/CF$) render it a promising electrocatalyst for sustainable $H_2$ production from water, ranking it the best among most of the reported nonprecious HER electrocatalysts (Supplementary Table 3 & Fig. 33)[29,60,61], including nanostructured $Ni_3N$, $Ni_3N/Ni(OH)_2$, and $Pt/Ni_3N$[42,62–64].

**Theoretical computations**. In order to shed light on the superior activity of $Ni_3N/Ni/NF$ as a HER electrocatalyst, DFT calculations were conducted on model systems. We modeled the $Ni_3N$ and blank Ni control samples by their lowest energy-surfaces of bulk $Ni_3N(001)$ and $Ni(111)$, respectively. In order to model the interfacing structure of $Ni_3N/Ni$, we reasoned that an appropriate structure was a few layers of nitrogen-terminated $Ni_3N$ located on the $Ni(111)$ surface. As proposed by Nørskov et al., the adsorption energy of hydrogen has been widely employed as a descriptor for predicting the HER performance of many electrocatalysts[3,27]. As shown in Fig. 3a and Supplementary Figs. 34–36, hydrogen atoms are preferred to adsorb along the interface between $Ni_3N$ and Ni in $Ni_3N/Ni$. In fact, two interfacial sites ($Ni_3N/Ni\_N$ and

$Ni_3N/Ni\_hollow$) were identified with very weak hydrogen adsorption energies (Supplementary Table 4). The resulting free energy changes ($\Delta G_{H^*}$) of hydrogen adsorption at these two positions of $Ni_3N/Ni$ (Fig. 3b) were calculated to be 0.01 and $-0.07$ eV, which are very close to 0 eV[27]. In contrast, pure Ni exhibits very strong hydrogen affinity with calculated $\Delta G_{H^*}$ close to $-0.30$ eV (Fig. 3b and Supplementary Figs. 37–38). Even though there exists one site on $Ni_3N$ which has a $\Delta G_{H^*}$ of 0.01 eV, there is another site on $Ni_3N$ showing very strong hydrogen binding affinity ($\Delta G_{H^*} = -0.57$ eV), which would not be beneficial towards efficient hydrogen electrochemistry (Fig. 3b and Supplementary Figs. 39–40).

In order to better understand the HER activity trend among $Ni_3N/Ni$, $Ni_3N$, and Ni under neutral and alkaline conditions which are lack of free protons, we sought to investigate the adsorption and dissociation of water on catalyst surface, which was believed to shed more lights on their hydrogen electrochemistry. The computed water adsorption energies on $Ni_3N/Ni$, $Ni_3N$, and Ni are compared in Fig. 3c and their corresponding adsorption configurations are shown in Supplementary Fig. 41. It is apparent that $Ni_3N/Ni$ and Ni possess similar water adsorption energies, much higher than that on $Ni_3N$. Based on the optimal water adsorption structure on $Ni_3N/Ni$, one can conclude that indeed the adsorbed water prefers to reside along the interface between $Ni_3N$ and Ni with one hydrogen atom pointing towards the edge N in $Ni_3N$, probably due to hydrogen bond interaction. Such interaction also facilitates the subsequent water dissociation on $Ni_3N/Ni$. Figure 3d and Supplementary Figs. 42–44 present the comparison of the activation energy barrier of water dissociation on $Ni_3N/Ni$, $Ni_3N$, and Ni. As expected, $Ni_3N/Ni$ shows the lowest energy barrier (0.50 eV) for water dissociation, nearly 0.08 eV smaller than that on $Ni_3N$ and 0.42 eV lower than that on Ni. The transition state structures of water dissociation on Ni, $Ni_3N$, and $Ni_3N/Ni$ are included in Fig. 3e. Collectively, these computational results further corroborate our hypothesis that the interfacial sites present on the surface of $Ni_3N/Ni$ indeed exhibit appropriate binding affinities towards hydrogen and water, and can also facilitate water dissociation, consistent with our experimentally obtained HER performance of $Ni_3N/Ni$ from neutral to alkaline conditions.

**Electrocatalytic $H_2$ oxidation**. The nearly zero catalytic onset potential for HER and very small free energy change of hydrogen adsorption of $Ni_3N/Ni/NF$ granted us confidence to believe that $Ni_3/Ni/NF$ could act as an excellent electrocatalyst for $H_2$ oxidation as well. Currently, it remains a critical challenge in developing inexpensive HOR electrocatalysts in alkaline electrolytes for the widespread employment of HEMFCs[6,9,65–67]. Due to the monolithic nature of the $Ni_3N/Ni/NF$ electrode, its HOR performance was measured in $H_2$-saturated 0.1 M KOH with continuous $H_2$ bubbling to mimic the HEMFC condition. For comparison, Pt/C was also loaded on NF and optimized (1.5 mg $cm^{-2}$) to achieve the best HOR activity under the similar conditions (Supplementary Fig. 45). The polarization curves of $Ni_3N/Ni/NF$ collected between 0 and 0.1 V vs. RHE in $H_2$ and Ar-saturated 0.1 M KOH are plotted in Supplementary Fig. 46. In contrast to the negligible capacitance current obtained in the Ar-saturated electrolyte, $Ni_3N/Ni/NF$ showed appreciable anodic current beyond 0 V vs. RHE upon $H_2$ saturation, implying the anodic current was due to $H_2$ oxidation. In fact, the HOR catalytic current of $Ni_3N/Ni/NF$ took off at 0 V vs. RHE, very close to that on Pt/NF, and surpassed the latter's at increasing applied potential (Fig. 4a). At 0.09 V vs. RHE, $Ni_3N/Ni/NF$ achieved a current density of 6.95 mA $cm^{-2}$, higher than that of Pt/NF (5.25 mA $cm^{-2}$), whereas Ni/NF only exhibited a current density of

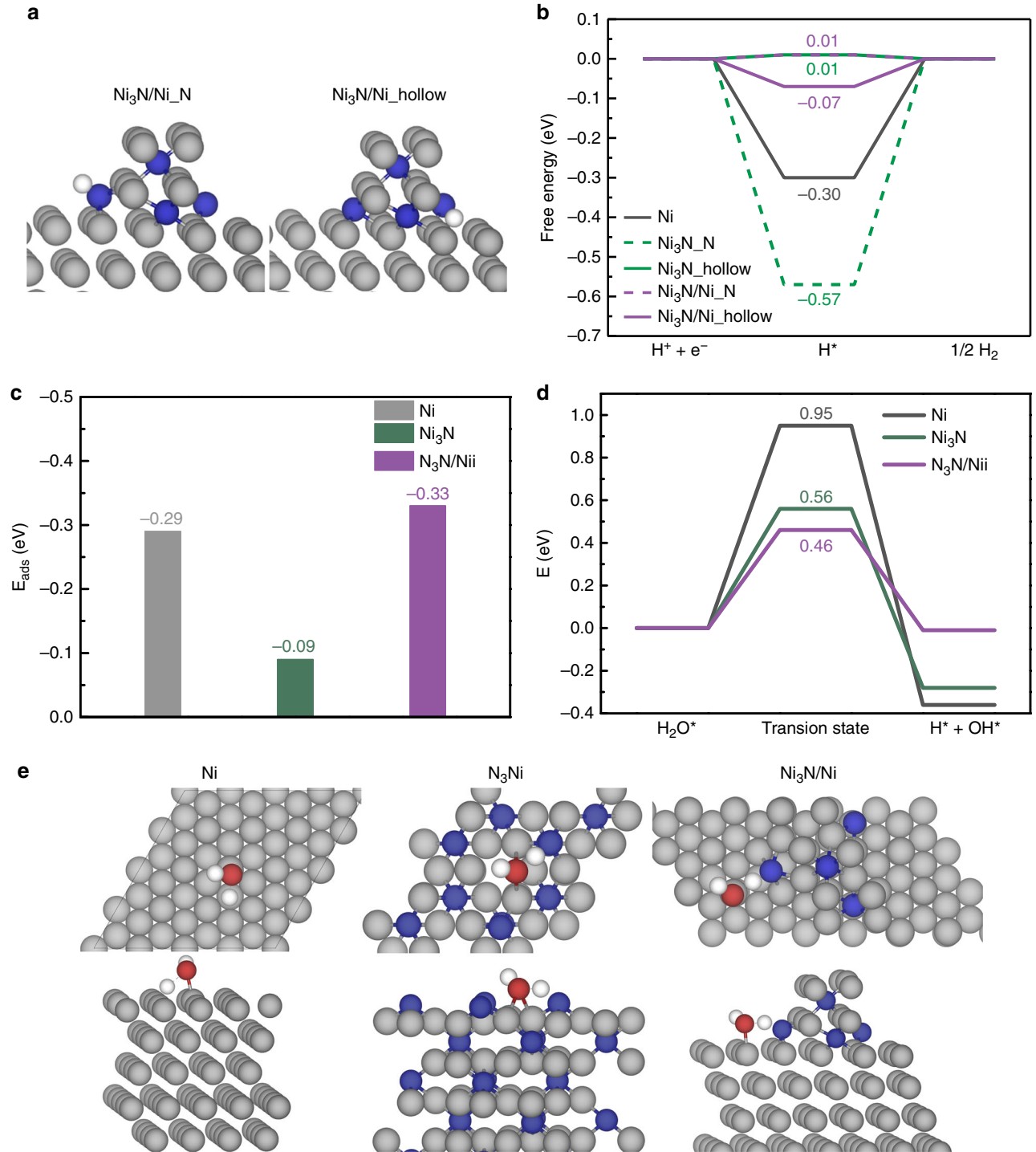

**Fig. 3** DFT calculation results. **a** DFT-optimized structures of hydrogen adsorption at two interfacial sites of Ni3N/Ni. **b** Hydrogen adsorption free energy ($\Delta G_{H^*}$) on Ni, Ni3N, and Ni3N/Ni. **c** Adsorption energy of water on Ni, Ni3N, and Ni3N/Ni. **d** Energy barrier for water dissociation on Ni, Ni3N, and Ni3N/Ni. **e** Transition state structures for water dissociation over Ni, Ni3N, and Ni3N/Ni, showing both the top view (top of **e**) and side view (bottom of **e**) of each structure. Color code: Ni: gray; N: blue; O: red; H: white

merely $0.34\ mA\ cm^{-2}$. The exchange current density was estimated from the micro-polarization region within a small potential window from $-20$ to $20\ mV$ vs. RHE (Supplementary Fig. 47). Both Ni3N/Ni/NF and Pt/NF showed electrocatalytic HER and HOR activities starting at nearly zero overpotential in 0.1 M KOH. The calculated exchange current density of Ni3N/Ni/NF was $3.08\ mA\ cm^{-2}$, which was 1.4 and 17.6 times that of Pt/

Ni and Ni/NF, respectively. It was found that Ni3N/Ni/NF showed higher specific activity than Pt/NF for HOR under similar experimental conditions, even if the current densities were normalized by their respective BET surface areas and/or ECSAs (Supplementary Figs. 48–49). The stability of Ni3N/Ni/NF for long-term $H_2$ oxidation was assessed via chronoamperometry at 0.09 V vs. RHE in $H_2$-saturated 0.1 M KOH and compared with

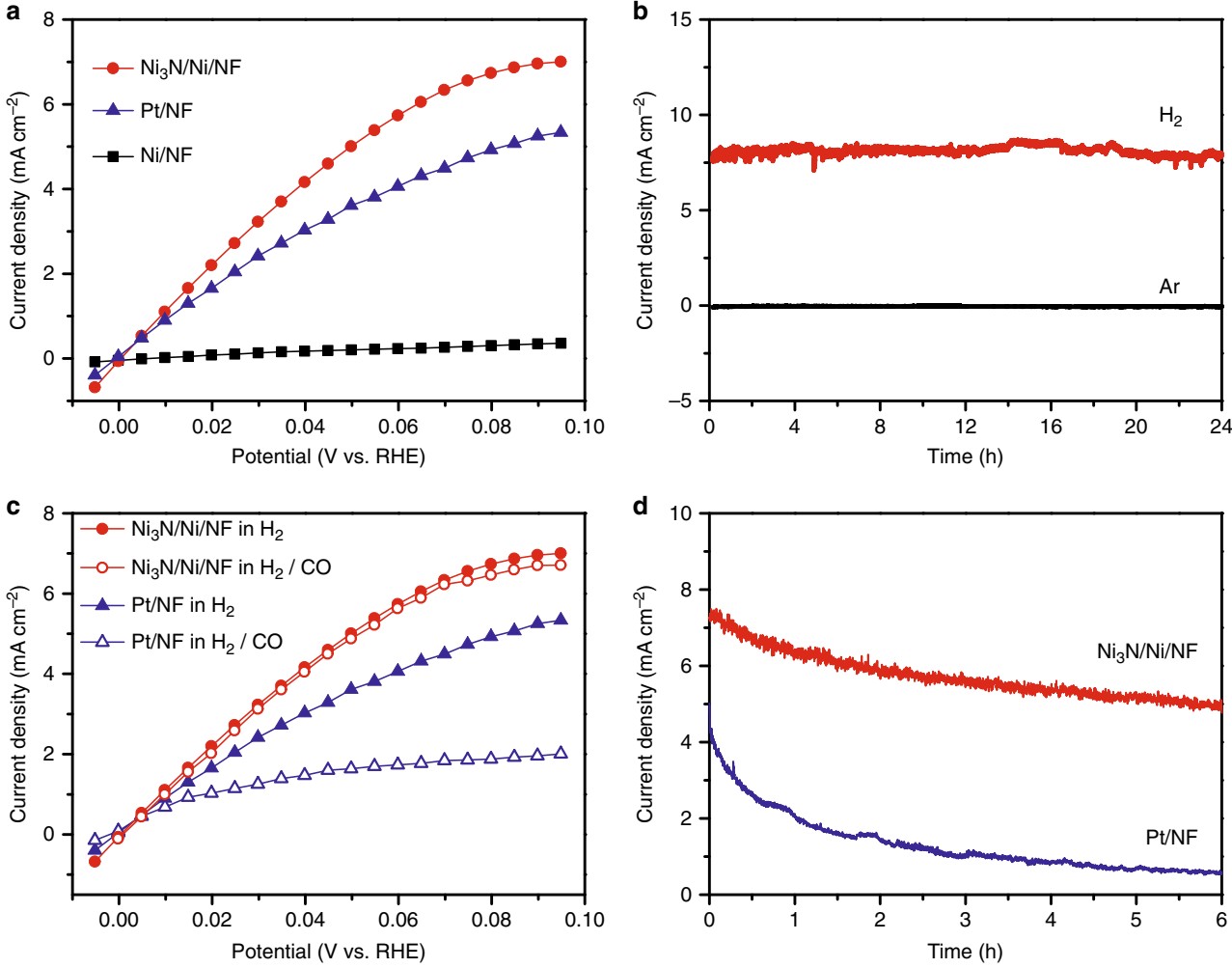

**Fig. 4** The electrocatalytic performance for HOR. **a** Steady-state polarization curves of Ni₃N/Ni/NF, Ni/NF, and Pt/NF for HOR in H₂-saturated 0.1 M KOH. **b** Chronoamperometry curves of Ni₃N/Ni/NF in H₂- or Ar-saturated 0.1 M KOH measured at 0.09 V vs. RHE. **c** Comparison of the HOR polarization curves of Ni₃N/Ni/NF and Pt/NF in 0.1 M KOH saturated with H₂ or H₂ with 2% CO (v/v). **d** Chronoamperometry curves of Ni₃N/Ni/NF and Pt/NF in 0.1 M KOH saturated with H₂ with 2% CO (v/v) measured at 0.09 V vs. RHE. All results were collected without iR compensation. The current densities were calculated on the basis of the geometric electrode areas

that collected in the absence of H₂ (Fig. 4b). Ni₃N/Ni/NF maintained a stable anodic current density of *ca.* 8 mA cm⁻² for 24 h when H₂ was bubbled through the electrolyte, while negligible current density was obtained in Ar-saturated 0.1 M KOH. In sharp contrast, if Ni/NF was utilized as the working electrode, no more than 0.5 mA cm⁻² could be obtained in H₂-saturated electrolyte (Supplementary Fig. 50).

In addition, the HOR performance of Ni₃N/Ni/CF and Ni₃N/NF was also measured under the same condition as Ni₃N/Ni/NF. Ni₃N/Ni/CF exhibited appreciable anodic current for HOR, although its activity and stability were lower than those of Ni₃N/Ni/NF, most likely due to the inferior porosity of carbon foam for H₂ diffusion (Supplementary Figs. 51–52). On the other hand, Ni₃N/NF only showed negligible anodic current for HOR, highlighting the advantages of utilizing rough and porous Ni/NF as the Ni source for the preparation of Ni₃N/Ni/NF (Supplementary Fig. 53). Since the hydrogen (H*) adsorption free energy ($\Delta G_{H^*}$) is also regarded as a key descriptor in determining HOR activity[6,8,9,67], the very small $\Delta G_{H^*}$ value (0.01 eV) of Ni₃N/Ni/NF mentioned above also supports its excellent HOR performance (Fig. 3b). In addition, we further evaluated the adsorption energies of H₂ on Ni₃N/Ni interface, Ni₃N, and Ni via DFT calculation (Supplementary Figs. 54–60). Ni₃N/Ni interface and Ni possess stronger H₂ adsorption than Ni₃N,

indicating favored H₂ adsorption on Ni₃N/Ni and Ni. Overall, given both its ideal $\Delta G_{H^*}$ value and strong H₂ adsorption, Ni₃N/Ni was DFT computationally predicted to be a great HOR electrocatalyst, in agreement with the aforementioned experimental results.

Since the current industrial production of H₂ mainly relies on steam reforming from hydrocarbons, which may result in CO impurity in the final H₂ gas. Therefore, high tolerance to CO impurity is a desirable property of electrocatalysts in hydrogen fuel cells. Unfortunately, CO poisoning is notoriously intolerable for Pt-based HOR electrocatalysts. Herein, we conducted CO tolerance tests in an extreme case wherein a H₂ gas mixture with 2% CO (v/v) was utilized. As shown in Fig. 4c, the HOR activity of Pt/NF indeed was strongly suppressed by the presence of CO, as its steady-state HOR polarization curves decreased substantially in the H₂/CO mixture relative to that in pure H₂. For instance, at 0.09 V vs. RHE, the maximum current density of Pt/NF decreased by ~63% from 5.25 to 1.95 mA cm⁻². On the other hand, Ni₃N/Ni/NF demonstrated a much better resistance towards CO poisoning, showing a much less decrease of its HOR polarization in the presence of CO. In fact, the maximum current density achieved at 0.09 V vs. RHE was only decreased by 3.5% on Ni₃N/Ni/NF, from 6.95 to 6.71 mA cm⁻². Next, chronoamperometry at 0.09 V vs. RHE was also conducted to

further evaluate their CO tolerance. As shown in Fig. 4d, a rapid current decrease was observed for Pt/NF in $H_2$/CO-saturated 0.1 M KOH, resulting in a merely 0.57 mA cm$^{-2}$ after 6 h electrolysis. In contrast, $Ni_3N$/Ni/NF was able to retain above 5 mA cm$^{-2}$ under the same conditions. These results unambiguously proved that our $Ni_3N$/Ni/NF showed exceptional CO tolerance for HOR, even though our testing conditions utilized a CO percentage at least two orders higher than those typically reported[68].

## Discussion

In summary, we have demonstrated that interfacing $Ni_3N$ and Ni on metallic nickel foam is an effective approach to producing highly active and robust electrocatalysts for both $H_2$ evolution and oxidation reactions in aqueous media. The resultant $Ni_3N$/Ni/NF catalyzes HER/HOR starting at zero overpotential, robust long-term durability, and great tolerance to CO poisoning. The superior electrocatalytic performance makes $Ni_3N$/Ni/NF the most active catalyst among most of the reported inexpensive electrocatalysts and it can even rival the activities of the state-of-the-art Pt/C catalysts loaded on NF under similar experimental conditions. A suite of physical characterizations, electrochemical experiments, together with theoretical computations, were conducted to gain the insights into the exceptional HER/HOR activities of $Ni_3N$/Ni/NF, which can be summarized in the following aspects.

The unique electronic and geometrical structures of the interfacial sites on $Ni_3N$/Ni provide great accommodation for hydrogen adsorption. As estimated from DFT calculations, the free energy change of hydrogen adsorption at the interfacial sites of $Ni_3N$/Ni/NF is very close to zero, which is beneficial to hydrogen electrochemistry[66]. Although fully considering the solvent environment and taking into account of water adsorption when calculating the hydrogen adsorption are challenging and beyond the scope of this work, our ongoing work aims to compute the apparent hydrogen adsorption energy, which has been recently proposed as a pH-dependent descriptor for HER and HOR activities[69].

Due to the lack of free protons in neutral and alkaline electrolytes, water adsorption, and dissociation have been proposed to be critical for HER at high pH. Our computational results suggest that the interface between $Ni_3N$ and Ni significantly promotes the initial water adsorption and reduces the energy barrier for the subsequent water dissociation compared to the situations on pure Ni or $Ni_3N$. The $Ni_3N$/Ni interface may also lower the energy barrier for the reorganization of the interfacial water network and enable efficient proton/hydroxide transfer through the double layer, thereby promoting the HER/HOR kinetics[72]. Another possible factor that cannot be completely excluded is the potential formation of nickel oxides/hydroxides on the surface of $Ni_3Ni$/Ni/NF during HER/HOR testing. Despite the debate over the promotional effect of interface oxophilicity[6,25,65–67,70,71], Markovic et al. proposed that regulating metal/metal (oxy)hydroxide interface can promote water dissociation for HER and optimize the balance between the active sites for $H_2$ adsorption/dissociation and the sites for hydroxyl adsorption, in order to enhance the alkaline HOR[30–33]. Analogous enhancement due to surface nickel oxides/hydroxides may also exist, however it should not be attributed as the primary factor, because the control sample Ni/NF, which should have similar tendency to form surface nickel oxides/hydroxide species, does not exhibit appreciable HER or HOR performance.

The intimate contact between the $Ni_3N$/Ni nanoparticles and the nickel foam substrate as well as the intrinsically metallic properties of both $Ni_3N$ and Ni enable fast electron transfer between the active sites and the current collector. The hierarchical topology and highly porous morphology of $Ni_3N$/Ni/NF not only maximize the accessibility of active sites but also facilitate mass transport, which are all beneficial to electrocatalytic $H_2$ evolution and oxidation reactions.

## Methods

**Syntheses of Ni/NF and $Ni_3N$/Ni/NF.** The $Ni_3N$/Ni/NF electrodes were prepared by cathodic electrodeposition of Ni particles on nickel foams followed by thermal nitridation. The electrodeposition was carried out with a two-electrode configuration in a cell containing $NH_4Cl$ (2.0 M) and $NiCl_2$ (0.1 M) at room temperature. A piece of clean nickel foam (0.5 cm×0.5 cm) and a carbon rod were used as the working and counter electrodes, respectively. The electrodeposition was performed at a constant current density of −1.0 A cm$^{-2}$ for 500 s under $N_2$ protection without stirring to obtain Ni/NF. Then the resultant Ni/NF was placed in the center of a quartz tube purged with $NH_3$ flow. It was heated to the desired temperature at a ramping rate of 10 °C min$^{-1}$ and maintained at the same temperature for a given duration. Finally, the furnace was naturally cooled down to room temperature, leading to $Ni_3N$/Ni/NF. The $NH_3$ flow was kept throughout the whole process. Two control samples of $Ni_3N$/Ni/CF and $Ni_3N$/NF were also synthesized under same conditions for comparison (see Supplementary Information).

**Electrocatalytic measurements.** The linear sweep voltammetry (LSV), cyclic voltammetry (CV), chronopotentiometry (CP), and chronoamperometry (CA) experiments were conducted using a Gamry Interface 1000 electrochemical workstation with a three-electrode configuration. The monolithic $Ni_3N$/Ni/NF was directly used as the working electrode. A calibrated Ag/AgCl (saturated KCl) with salt bridge kit and a carbon rod were used as the reference and counter electrode, respectively. The electrolyte for HER was 1.0 M potassium phosphate buffer (KPi, pH 7.17), or 1.0 M KOH (pH 13.80). The electrolyte for HOR was 0.1 M KOH (pH 12.80). All electrolytes were bubbled with $H_2$ throughout the whole electrochemical experiments. All potentials are reported versus reversible hydrogen electrode (RHE) according to the following equation:

$$E(\text{vs.RHE}) = E(\text{vs.Ag/AgCl}) + 0.197 + 0.059 \times \text{pH} \qquad (1)$$

Hg/HgO (1.0 M KOH, CH Instruments) and $Hg/Hg_2SO_4$ (saturated $K_2SO_4$, CH Instruments) reference electrodes were also used to verify the electrocatalytic performances which were consistent with the results referenced to Ag/AgCl (saturated KCl) electrodes. The LSV and CV curves were collected at 5 mV s$^{-1}$. Unless stated otherwise, all LSV polarization curves for HER were iR-corrected and obtained by scanning from negative to positive potential. The correction was made according to the following equation:

$$E_{\text{corrected}} = E_{\text{measured}} - iR_s \qquad (2)$$

where $E_{\text{corrected}}$ is the iR-corrected potential, $E_{\text{measured}}$ and $i$ are experimentally measured potential and current, respectively, and $R_s$ is the equivalent series resistance measured via electrochemical impedance spectroscopy in the frequency range of 10$^6$–0.1 Hz with an amplitude of 10 mV.

For HOR tests, the steady-state measurements were conducted to obtain the polarization curves instead of LSV or CV methods to minimize the capacitive current background. The multi-step CA was conducted at a potential window from −0.05 to 0.1 V vs. RHE with a 5 mV interval for every 60 s. The stable anodic current recorded at 60 s under each potential was used to plot the steady-state polarization curves for HOR. For comparison, the polarization curves towards HOR were also collected in 0.1 M KOH bubbled with high-purity $H_2$ or $H_2$ containing 2% CO (v/v). The catalytic stability for HER/HOR was evaluated by either CP or CA measurement without iR correction. Besides $Ni_3N$/Ni/NF, the electrodeposited Ni/NF, $Ni_3N$/Ni/CF, $Ni_3N$/NF and commercial Pt/C catalysts loaded on nickel foams with optimized loading were also used as working electrodes for both HER and HOR. For HER, the optimal loading of Pt/C powder on nickel foam (Pt/NF) was 2.5 mg cm$^{-2}$ in 1.0 M KPi and 1.0 M KOH. For HOR in 0.1 M KOH, the optimal loading of Pt/C on nickel foam (Pt/NF) was 1.5 mg cm$^{-2}$. The current densities in this work were calculated on the basis of the geometric areas, BET surface areas, or electrochemically active surface areas (ECSAs) of electrodes.

The exchange current ($i_0$) can be obtained by fitting kinetic current ($i_k$) versus the overpotential ($\eta$) using the Butler–Volmer Eq. (3),

$$i_k = i_0 \left( e^{\frac{\alpha F}{RT}\eta} - e^{\frac{(\alpha-1)F}{RT}\eta} \right) \qquad (3)$$

where $\alpha$ is the charge transfer coefficient, $\eta$ is the overpotential, $R$ is the ideal gas constant (8.314 J mol$^{-1}$ K$^{-1}$), $T$ is the experimental temperature (298 K), and $F$ is the Faradaic constant (96,485 C mol$^{-1}$).

In a small potential window of the micro-polarization region near the equilibrium potential (±20 mV vs. RHE), $i_k$ approximately equals to the measured current ($i$). In this case, the Butler–Volmer equation can be expanded by Taylor's

formula and simplified as Eq. (4),

$$i = i_0 \frac{\eta F}{RT} \qquad (4)$$

Therefore, $i_0$ can be obtained from the slope of the linear fitting in the micro-polarization region[5,25,72]. The exchange current density ($j_0$) was calculated by dividing $i_0$ by the geometric electrode area.

**Theoretical computation.** The DFT calculations were performed with Vienna Ab initio Simulation Package (VASP) version 5.4[73–76]. The projector augmented-wave (PAW) potentials[77] were used, with 1 s of H, 2s2p of N and O, and 3d4s of Ni treated as valence electrons. The generalized gradient approximation (GGA) of Perdew, Burke, and Ernzerhof (PBE)[78] was employed. A cutoff energy of 450 eV was used for the plane-wave basis set. The Brillouin zone was sampled by Monkhorst-Pack k-point mesh, with reciprocal lattice spacing ≤0.04 Å$^{-1}$. These settings converge the total energy to ≤1 meV/atom with respect to higher kinetic energy cutoff or denser k-point mesh. The convergence criterion for structural optimization was set to 0.025 eV/Å for each atom.

The optimized bulk face centered cubic Ni structure has a lattice constant of 3.515 Å (experimental lattice constant is equal to 3.523 Å[79]. The lattice constants for optimized bulk hexagonal $Ni_3N$ are $a = 4.612$ Å, $c = 4.302$ Å (experimental values: $a = 4.622$ Å and $c = 4.306$ Å[79]). Predicted lattice constants for both materials deviate <0.3 % from their corresponding experimental values. The optimized bulk structures were used to construct surface slab models. A ($4 \times 4$) Ni (111) slab model of five layers was used for pure Ni, with the bottom two layers of Ni atoms fixed to mimic bulk structure. For $Ni_3N$, a N-terminated ($2 \times 2$) $Ni_3N$ (001) surface slab of 10 layers was built, with the bottom four layers fixed . For the hybrid model, a four-layer ($6 \times 3$) Ni (111) slab was constructed, with the bottom two layers fixed and an $Ni_3N$ nanowire placed on top. The nanowire consists of four layers of $Ni_3N$ (001), with N termination interacting with Ni (111) surface to make an effective interface. While both Ni layers in Ni (111) and $Ni_3N$ (001) form hexagonal array, the nearest Ni–Ni distance in $Ni_3N$ (001) planes is ~8% longer than its counterpart in Ni (111). Therefore, the $Ni_3N$ nanowire in the hybrid model experiences compressive strain from Ni substrate. The two sides of the nanowire correspond to N-terminated (110) surface of $Ni_3N$.

The adsorption energy of hydrogen was defined as $E_{slab-H} - \left( E_{slab} + \frac{1}{2} E_{H_2} \right)$. Zero-point energy and entropic corrections were included for calculating the Gibbs free energy correlations (see Supplementary Information for more details). The reaction pathway was simulated by the climbing image nudged elastic band (CI-NEB) [80] and the dimer[81] method.

## Data availability

The data that support the findings of this study are available from the corresponding authors on reasonable request.

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

## Acknowledgements

This work was supported by the UTAG program of USTAR, State of Utah. Y.S. acknowledges the financial support of Herman Frasch Foundation (820-HF17), National Science Foundation (CHE-1653978), and the University of Cincinnati. We acknowledge the Microscopy Core Facility at USU. P.L. gratefully acknowledges the American Chemical Society Petroleum Research Fund for support of this research. The computational research was supported in part through computational resources provided by the Information Technology department at Purdue, West Lafayette, Indiana.

## Author contributions

Y.S. designed and supervised the project, directed the research, analyzed and interpreted the data, and wrote the manuscript. F.S. and W.L. designed the methodology and conducted the experiments, synthesized samples, analyzed and interpreted the data, and wrote the manuscript. F.S. and W.L. contributed equally to this work. G.H. contributed to SEM and elemental mapping measurements. P.L. and J.Q. contributed to DFT computations. All of the authors discussed the results and reviewed the manuscript.

## Additional information

**Competing interests:** The authors declare no competing interests.

