## [Peer Review File · Nature Communications]

Reviewers' comments:

Reviewer #1 (Remarks to the Author):

This manuscript shows that interfacing Ni and Ni₃N results in an electrocatalyst with extraordinary activities for both HER and HOR in a broad pH range. However, the novelty of this work cannot meet the high standards of Nature Communications and I must reject it.

(1) The author mentioned that the rich Ni₃N/Ni interfacial sites can be obtained by electrodeposition of Ni nanoparticles on nickel foams followed by nitridation in ammonia. This preparation process is similar to reported work (J. Am. Chem. Soc. 2017, 139, 12283–12290). Also, Ni₃N for HER electrocatalysis is also not new.

(2) Please provide the reason of electrodeposition Ni microsphere as Ni source rather than using Ni foam directly. Please discuss the function of Ni foam. Can carbon cloth replace Ni foam?

(3) The author claimed that Ni₃N/Ni/NF demonstrates unprecedented HER activity with nearly zero onset overpotential from pH 0 to 14. This is a surprising result as Ni is known to be soluble in acid solution.

(4) The author mentioned that the current densities are calculated on the basis of the geometric area of each electrode. For such 3D catalyst, geometric area cannot truly reflect the real area of the material and specific area normalized current densities should be provided to better reflect the intrinsic activity of Ni₃N/Ni/NF.

(5) In Fig. 2a and 2c, the current densities is positive when the potential is zero vs. RHE. Please explain it. Please offer the HER performance of Ni₃N/NF.

Reviewer #2 (Remarks to the Author):

This manuscript reports a plausible Ni₃N/Ni electrocatalyst with certain HER/HOR performance in aqueous electrolytes at different pH values, in combination with density functional theory computations. The claims are pretty interesting and if the claims were correct and convincing, they would be nice contributions to a wide community. My concerns are listed in the following:

I am not sure if the interface of Ni₃N/Ni does exist and is available for HER/HOR merely based on SEM and TEM images. Detailed structural characterizations, such as high-resolution (sub-nanometer) elemental mapping, are required to investigate the formation of Ni₃N/Ni and to quantify the coverage percentage of Ni₃N on the Ni substrate. In addition, is it possible to experimentally determine the thickness of Ni₃N?

Ag/AgCl was employed as the reference electrode in this study. However, the authors did not specify if a salt bridge was used to separate Ag/AgCl from the electrolyte. If not, chloride would seriously poison the surface of platinum, thus underestimating the electrochemical performance of platinum and losing a fair comparison. In this case, all of the claims about electrochemical performance will not make sense.

It is critical to use the best performance of platinum as the standard, so the data on the optimization of Pt/C toward HER/HOR on Ni foam are required.

I cannot tell the HOR part is a good addition or else, since no DFT study was carried out for HOR.

Normally, the performance of Pt/C becomes lower and lower with the increase of pH values, why Pt/C in this study did not follow such a trend?

The same references are repetitively listed and should be corrected.

Is the labeling of supplemental Figure 14 correct?

For HER plots, if iR-correction has been done?

Reviewer #3 (Remarks to the Author):

In this work the authors report on the fabrication of Ni₃N/Ni foam electrode as electrocatalyst for hydrogen evolution and oxidation reaction. The material and the synthesis are not new but the performance and mechanistic insights are very interesting. Therefore, I recommend its publication after addressing the following points:

1. Many works show the direct growth of Ni₃N on nickel foam as electrocatalyst for HER/OER. In these works, also they can obtain the Ni/Ni₃N structure. The authors should compare their results to a direct Ni₃N growth. Moreover, as they create high surface area network Ni₃N particles - a comparison of the Ni₃N on carbon foam is required to prove the synergetic effect with Ni foam.
2. Can they show if they control the Ni₃N thickness/amount and how does it affect the catalytic properties?
3. The authors should provide some post-characterizations as XPS, XRD, etc. to further support the high stability (especially in acid).
4. The authors should further elaborate the Tafel slope values in KOH. What is the mechanism for the Ni₃N/Ni?
5. the theoretical part is very interesting - how is the mechanism compared to Pt electrode? In addition, it is still not really clear why a Tafel mechanism is observed in 1M KOH. Due to the nitrogen sites?

Point-by-Point Response Letter

Referee # 1:

This manuscript shows that interfacing Ni and Ni₃N results in an electrocatalyst with extraordinary activities for both HER and HOR in a broad pH range. However, the novelty of this work cannot meet the high standards of Nature Communications and I must reject it.

Our response: We respectfully disagree with the reviewer's conclusion that "the novelty of this work cannot meet the high standards of Nature Communications". We have provided more data and evidence to experimentally support the exceptional activities of Ni₃N/Ni/NF for both HER and HOR and to shed light on the insights into such unprecedented performance by density functional theory (DFT) calculations in the revised manuscript. Herein, we would like to highlight the novelty and significance of our work beyond the state of the art.

Although many nanostructured Ni₃N and composites have been developed through sophisticated synthetic methods and used as electrocatalysts for HER, their performances in those reports are still much inferior to those of Pt catalysts (**Supplementary Table 3**). **To the best of our knowledge, no catalytic systems ever reported focus on exploring the interfaces of 1st-row transition metals and their nitrides for hydrogen electrochemistry (HER and HOR) in aqueous media**, even though some Ni₃N-based composites have been reported for catalyzing HER in either neutral or alkaline solution, such as Ni₃N/carbonaceous substrates (*J. Mater. Chem. A*, 2017, 5, 9377–9390; *J. Mater. Chem. A*, 2016, 4, 17363–17369; *Inorg. Chem. Front.*, 2017, 4, 1120–1124; *J. Mater. Chem. A*, 2015, 3, 8171–8177), Ni₃N/nickel borate (*J. Mater. Chem. A*, 2017, 5, 7806–7810), Ni₃N/TiN (*J. Mater. Chem. A*, 2016, 4, 5713–5718), Ni₃N/NiMoN (*Nano Energy*, 2018, 44, 353–363) and Ni₃N/Pt (*Adv. Energy Mater.*, 2017, 7, 1601390). In fact, the catalytic activity of Ni₃N towards HOR still remains unexplored. None of the above publications comprehensively investigate the effects of transition metal/transition metal nitride interfaces on some critical properties governing the H₂ electrochemistry, including hydrogen adsorption free energy (ΔG_{H^*}), adsorption energy of water, energy barrier of water dissociation and adsorption energy of H₂.

Our work demonstrates that rationally interfacing Ni and Ni₃N results in a versatile electrocatalyst (Ni₃N/Ni) with extraordinary activities for both HER in neutral and alkaline electrolytes and HOR in alkaline solution. The novelty and significance of our work are shown as follows.

(i) The abundant Ni₃N/Ni interfacial sites are fabricated by facile electrodeposition of porous Ni particles on common current collectors of Ni foam or carbon foam, followed by controllable thermal nitridation in NH₃. The simple cathodic electrodeposition method endows the electrocatalyst with hierarchical porous architecture and large surface area, while the amount of Ni₃N can be readily tuned by adjusting thermal nitridation time. **The electrodeposition and nitridation operations can be carried out under atmospheric pressure in a facile and scalable manner which is favorable for large-scale industrial production, whereas fabrication of Ni₃N composite catalysts usually involves sophisticated high-pressure hydrothermal/solvothermal procedures (see references above).**

(ii) The rich Ni₃N/Ni interfacial sites offer extraordinary electrocatalytic activities for both HER and HOR, **which can unambiguously rival those of Pt electrocatalysts in terms of both apparent activities (normalized by the geometric electrode areas) and intrinsic specific activities (normalized by the BET and electrochemically active surface areas) together with remarkable durability.** The performances of such Ni₃N/Ni electrocatalyst towards HER in neutral and alkaline solutions are substantially better than those of reported Ni₃N and Ni₃N-based composite electrocatalysts (**Supplementary Table 3**).

(iii) Moreover, the **HOR performance of such Ni₃N/Ni electrocatalyst in 0.1 M KOH is also remarkable**, demonstrating zero overpotential of catalytic onset, robust long-term stability and outstanding CO tolerance, **which has never been reported for nonprecious electrocatalysts to the best of our knowledge.**

(iv) Finally, **our DFT computation results elucidate the effects of Ni₃N/Ni interface on some crucial properties regulating the hydrogen electrochemistry, encompassing ΔG_{H^*} , adsorption energy of H₂O, energy barrier of water dissociation and adsorption energy of H₂**, which will provide guidance for the rational design of inexpensive HER electrocatalysts and advance the development of nonprecious HOR electrocatalysts.

Therefore, we believe that the novelty and significance of our work meet the high standards of *Nature Communications* very well. We sincerely hope the reviewer would kindly reconsider the revised manuscript that reflects the aforementioned novelty and significance.

(1) The author mentioned that the rich Ni₃N/Ni interfacial sites can be obtained by electrodeposition of Ni nanoparticles on nickel foams followed by nitridation in ammonia. This preparation process is similar to reported work (*J. Am. Chem. Soc.* 2017, 139, 12283–12290). Also, Ni₃N for HER electrocatalysis is also not new.

Our response: We sincerely appreciate the reviewer's comment and his/her mention of our previous work (*J. Am. Chem. Soc.*, 2017, 139, 12283–12290). However, we have to regretfully point out that the reviewer misunderstood (or missed) the main selling point of our current work. Our previous JACS paper reported a surface nitrogen modification strategy to improve the HER performance of Ni (N-Ni) in the neutral solution through ammonium carbonate treatment. It was found that the ammonium carbonate treatment only introduced surface nitrogen which has a very weak interaction with metallic Ni and did not significantly change the chemical composition and phase of the underlying Ni framework. **The comprehensive characterization results (e.g., XAS, XRD, XPS, and TEM) of N-Ni in that report validate the surface nitrogen modification of Ni instead of the formation of nickel nitride or Ni₃N/Ni interfaces.** In sharp contrast, our present work focuses on developing **crystalline Ni₃N/Ni interfaces** by NH₃ treatment. The co-existence of crystalline Ni₃N and Ni can be clearly seen in the XRD pattern of the Ni₃N/Ni catalyst on carbon foam (Ni₃N/Ni/CF, **Figure R1a**) and on Ni foam (Ni₃N/Ni/NF, **Figure 1** in the main text). The Ni₃N/Ni interface is also demonstrated in the TEM image (Figure R1b) and elemental maps (**Figure R2**).

Figure R1. (a) XRD pattern of Ni₃N/Ni/CF. (b) HRTEM image of Ni₃N/Ni interface.

Figure R2. HAADF-STEM image and corresponding elemental maps of Ni and N in $\text{Ni}_3\text{N}/\text{Ni}$ interfacial electrocatalyst. Scale bar, 2.5 nm.

The surface nitrogen modification achieved by ammonium carbonate treatment represents a surface heteroatom modification strategy to promote HER. The introduction of surface nitrogen can effectively enhance the adsorption energy of H_2O and reduces the energy barrier for water dissociation, however, it cannot render hydrogen adsorption free energy (ΔG_{H^*}) close to zero. On the contrary, that reported work confirmed that the introduction of surface nitrogen (N-Ni) increases the value of ΔG_{H^*} compared to that of bare Ni. Therefore, **the surface nitrogen modification strategy** (*J. Am. Chem. Soc.*, 2017, 139, 12283–12290) **is proved to be only effective to enhance the HER performances of Ni to a limited extent under neutral and alkaline conditions.** Furthermore, we have tested the HOR performance of N-Ni (synthesized by the reported method in *J. Am. Chem. Soc.*, 2017, 139, 12283–12290) in 0.1 M KOH under the similar electrochemical characterization method to that of $\text{Ni}_3\text{N}/\text{Ni}$. It is found that the surface nitrogen modification of Ni cannot render N-Ni appreciable HOR performance (**Figure R3**), possibly due to the failure of lowering ΔG_{H^*} to zero through this manner.

Figure R3. Chronoamperometry curves of $\text{Ni}_3\text{N}/\text{Ni}/\text{NF}$ and N-Ni (synthesized according to *J. Am. Chem. Soc.*, 2017, 139, 12283–12290) for HOR at 0.09 V vs. RHE in H_2 -saturated 0.1 M KOH.

However, our current work aims to exemplify the feasibility of the proposed interfacing strategy to obtain a high-performance and versatile electrocatalyst for both HER and HOR and to reveal how the metal nitride/metal interface influences the H_2 electrochemistry by fabricating an electrocatalyst with rich $\text{Ni}_3\text{N}/\text{Ni}$ interfaces.

In contrast to the surface nitrogen modification strategy, the present work demonstrates that rationally interfacial Ni and crystalline Ni₃N results in a versatile electrocatalyst (Ni₃N/Ni) with extraordinary activities for both HER and HOR, representing a powerful interfacial strategy. **Interfacial Ni₃N with Ni not only effectively increases the adsorption energy of H₂O and lowers the energy barrier for water dissociation, but also makes the hydrogen adsorption free energy (ΔG_{H^*}) very close to zero.** These synergistic effects achieved by interfacial Ni and crystalline Ni₃N together with the structural advantages impart the Ni₃N/Ni electrocatalyst with exceptional activities for HER and HOR, significantly higher than those for N-Ni synthesized by the reported surface nitrogen modification strategy (**Supplementary Table 3 and Figure R4**). These results demonstrate that **the interfacial Ni₃N/Ni strategy is effective to substantially promote both HER (in neutral and alkaline solutions) and HOR (in alkaline solution).**

Therefore, the interfacial Ni₃N/Ni strategy in this work is **distinct from** the reported surface nitrogen modification strategy (*J. Am. Chem. Soc.*, 2017, 139, 12283–12290) and **demonstrates substantial advantages beyond the report in terms of composition, electrocatalytic activities and theoretical descriptors.**

Ni₃N and its composites have been indeed studied as HER electrocatalysts, as mentioned in the above discussion. Most reports employed high-pressure hydrothermal reaction combined with post-ammonia treatment. The fabrication method in this manuscript can be conducted **under atmospheric pressure in a facile and scalable manner which is favorable for large-scale industrial production.** Few previous Ni₃N works have investigated its HOR performance under alkaline conditions. None of the publications comprehensively probed the effects of transition metal/transition metal nitride interfaces on some critical properties governing the H₂ electrochemistry, including hydrogen adsorption free energy (ΔG_{H^*}), adsorption energy of water, energy barrier of water dissociation and adsorption energy of H₂. We have compared the HER performance of Ni₃N/Ni electrocatalyst with those of reported Ni₃N-based catalysts (**Supplementary Table 3**). The overpotential under given current density of Ni₃N/Ni electrocatalyst are smaller than those of reported Ni₃N-based catalysts. **In this work, the interfacial strategy imparts the Ni₃N electrocatalyst with exceptional performance and versatility for catalyzing both HER and HOR. Furthermore, our DFT computations reveal how the Ni₃N/Ni interface influences the key underlying descriptors regulating the H₂ electrochemistry.** Therefore, we believe that our work has significant novelty and advance compared to our previous work and believe it's suitable for Nature Communications. In the revised manuscript, we have highlighted the significance and novelty of this work.

(2) Please provide the reason of electrodeposition Ni microsphere as Ni source rather than using Ni foam directly. Please discuss the function of Ni foam. Can carbon cloth replace Ni foam?

Our response: The electrodeposited Ni microspheres on a Ni foam (Ni/NF) instead of a pristine Ni foam (NF) were used as the Ni source for the subsequent thermal nitridation, because **Ni/NF has rougher surface** composed of loosely stacked coarse Ni microparticles and hierarchical porous structure (arising from vigorous H₂ bubbling accompanied with the cathodic electrodeposition of Ni microspheres at a high current density of -1.0 A cm^{-2}), while a pristine Ni foam has a smooth Ni ligament surface.

As a control experiment, a pristine Ni foam (NF) was also directly nitridized in NH₃ (Ni₃N/NF) under the same condition as the preparation of Ni₃N/Ni/NF. The SEM image of Ni₃N/NF (**Figure R4**) confirms its smooth surface after thermal nitridation. In contrast, Ni₃N/Ni/NF possesses macroporous ligament network structure with numerous stacked coarse particles over the skeleton surface (**Figure 1a** in the main text). The comparison of the nitrogen adsorption-desorption isotherms (**Figure R5**) shows that Ni₃N/Ni/NF has a larger BET surface area than Ni₃N/NF. The electrocatalytic HER activities of Ni₃N/NF in alkaline and neutral solutions as well as its HOR performance in 0.1 M KOH are very much inferior to those of Ni₃N/Ni/NF (**Figure R6**). These results indicate the advantages of using Ni/NF as Ni source.

Therefore, we utilized electrodeposited Ni microspheres on the Ni foam (Ni/NF) as the Ni source for thermal NH_3 treatment rather than directly utilizing a bare Ni foam.

Figure R4. (a-c) SEM images and (d) elemental maps of $\text{Ni}_3\text{N}/\text{NF}$.

Figure R5. N_2 adsorption-desorption isotherms of $\text{Ni}_3\text{N}/\text{Ni}/\text{NF}$ and $\text{Ni}_3\text{N}/\text{NF}$.

Figure R6. Linear sweep voltammetry (LSV) curves of Ni₃N/Ni/NF and Ni₃N/NF for HER in (a) 1.0 M KOH and (b) 1.0 M KPi, and (c) chronoamperometry (CA) curves of Ni₃N/Ni/NF and Ni₃N/NF for HOR in H₂-saturated 0.1 M KOH measured at 0.09 V vs. RHE. The LSV curves for HER were iR-corrected, while the CA curves for HOR were obtained without iR-correction.

It is proposed that the Ni foam serves as a robust current collector and support substrate for Ni₃N/Ni interfacial catalytic sites for electrocatalytic HER and HOR and acts as a favorable macroporous gas diffusion medium for facilitating H₂ gas diffusion in the HOR.

In fact, our synthetic strategy is not limited to Ni foam. Following the reviewer's suggestions, we also utilized carbon foam (CF) as the substrate. Ni₃N/Ni interfaces could be successfully fabricated on a carbon foam to obtain Ni₃N/Ni/CF with the same synthetic approach except replacing NF with CF. The Ni₃N/Ni/CF was characterized by XRD (Figure R1a), SEM (Figure R7), and XPS (Figure R8), confirming the presence of Ni₃N/Ni interfaces over CF with the similar composition and morphology to those of Ni₃N/Ni/NF.

Figure R7. SEM images (a-c) and elemental maps (d) of Ni₃N/Ni/CF.

Figure R8. Comparison of (a) Ni 2p_{3/2} and (b) N 1s XPS spectra of Ni₃N/Ni/CF and Ni₃N/Ni/NF.

The HER performance of Ni₃N/Ni/CF was measured in 1.0 M KOH and 1.0 M KPi and compared with that of Ni₃N/Ni/NF. The linear sweep voltammetry (LSV) and chronopotentiometry (CP) curves of Ni₃N/Ni/CF almost overlap with those of Ni₃N/Ni/NF under the same conditions (**Figures R9**), demonstrating that the Ni₃N/Ni interfacial sites on CF have similar catalytic activity and stability as those on NF. **This indicates that NF only acts as a current collector and catalyst support with the same function as CF in the process of electrocatalytic HER.**

Figure R9. Linear sweep voltammetry curves of Ni₃N/Ni/NF and Ni₃N/Ni/CF in 1.0 M KOH (a) and 1.0 M KPi (b). The inset shows their chronopotentiometry curves measured at -10 mA cm^{-2} .

The HOR performance of Ni₃N/Ni/CF was also measured under the same condition as for Ni₃N/Ni/NF. In contrast to the negligible current obtained in the Ar-saturated 0.1 M KOH, Ni₃N/Ni/CF showed appreciable anodic current beyond 0 V vs. RHE upon H₂ saturation (**Figure R10a**), suggesting that Ni₃N/Ni/CF also has high catalytic activity towards HOR. The long-term stability of Ni₃N/Ni/CF for HOR was assessed by chronoamperometry at 0.09 V vs. RHE in H₂-saturated 0.1 M KOH (**Figure R10b**). The initial HOR current density of Ni₃N/Ni/CF was 3.64 mA cm⁻² and gradually decreased to 2.72 mA cm⁻² after 24 h. Note that the HOR current density of Ni₃N/Ni/CF was lower than that of Ni₃N/Ni/NF under the same condition, indicating the promotional impact of Ni foam. Ni foam can serve as a favorable H₂ gas diffusion layer besides the roles of current collector and catalyst support. HOR occurs at a three-phase interface (gaseous H₂–liquid KOH electrolyte–solid Ni₃N/Ni catalyst), in which H₂ diffusion plays an important role in the measured current (*Nature Commun.*, 2014, 6, 5848; *Sci. China Mater.*, 2016, 59, 217–238; *Adv. Mater.*, 2017, 29, 1604685). Therefore, gas diffusion greatly influences the measured HOR current. Ni foam has a 3D macroporous network structure with better porosity and mechanical robustness than a carbon foam composed of interconnected carbon fibers for H₂ diffusion (**Figure R7**). It was reported that metal-based gas diffusion layers have advantages over carbon-based diffusion layers in gas electrodes (*Adv. Mater.*, 2017, 29, 1604685). In contrast, HER takes place at a two-phase interface (liquid electrolyte–solid Ni₃N/Ni catalyst), wherein liquid electrolyte is abundant, making diffusional component of the measured current almost negligible (*J. Electrochem. Soc.*, 2010, 157, B1529–B1536; *Sci. China Mater.*, 2016, 59, 217–238). Consequently, Ni₃N/Ni/CF and Ni₃N/Ni/NF showed similar HER performance. **In summary, Ni foam serves as a robust current collector and catalyst support for electrocatalytic HER and HOR and also acts as a macroporous gas diffusion medium for facilitating H₂ gas diffusion in HOR. Ni foam can be replaced with carbon foam, leading to similar HER performance but slightly lower HOR performance.**

Figure R10. (a) Steady-state polarization curves of Ni₃N/Ni/CF in 0.1 M KOH saturated with H₂ or Ar. (b) Chronoamperometry curves of Ni₃N/Ni/CF in 0.1 M KOH saturated with H₂ or Ar measured at 0.09 V vs. RHE. All results were collected without iR compensation.

(3) The author claimed that Ni₃N/Ni/NF demonstrates unprecedented HER activity with nearly zero onset overpotential from pH 0 to 14. This is a surprising result as Ni is known to be soluble in acid solution.

Our response: We confirm that Ni₃N/Ni/NF demonstrates unprecedented HER activity with nearly zero onset overpotential from pH 7 to 14. We also found that Ni₃N/Ni/NF showed remarkable HER activity in 0.5 M H₂SO₄ during the short duration of electrochemical testing, even though Ni might gradually dissolve in acidic solution. Actually, Ni foams coated with various *in situ* grown transition metal-based electrocatalysts such as Ni_{1-x}Co_xSe₂ (*Adv. Mater.*, 2017, 29, 1606521), Ni₃S₂ (*J. Am. Chem. Soc.*, 2015, 137, 14023–14026), Co-Ni-P (*J. Mater. Chem. A*, 2016, 4, 10195–10202; *Catal. Today*, 2017, 287, 122–129) and Ni₂P (*ACS Appl. Mater. Interfaces*, 2015, 7, 2376–2384) have been reported to show good HER performance in 0.5 M H₂SO₄ within limited duration of electrochemical testing. It is speculated that the coating layer with high HER activity and relatively good resistivity against acidic corrosion may slow

down (if not completely prohibit) the dissolution of Ni foam. However, considering the limited long-term stability of Ni foam at pH 0, the electrocatalytic HER study of Ni₃N/Ni/NF in the acidic electrolyte has been removed and we focus on emphasizing the unprecedented HER activity of Ni₃N/Ni/NF in the neutral and alkaline electrolytes in the revised manuscript.

(4) The author mentioned that the current densities are calculated on the basis of the geometric area of each electrode. For such 3D catalyst, geometric area cannot truly reflect the real area of the material and specific area normalized current densities should be provided to better reflect the intrinsic activity of Ni₃N/Ni/NF.

Our response: We agree with the reviewer that the geometric area for 3D catalysts cannot truly reflect the real area of the materials (*Joule*, 2018, 2, 1024-1027). In order to assess the intrinsic specific activity, the electrocatalytic activities of Ni₃N/Ni/NF are normalized by the Brunauer-Emmett-Teller (BET) surface area measured by N₂ adsorption-desorption and the electrochemically active surface area (ECSA) measured by the double layer capacitance method on the basis of cyclic voltammetry in nonaqueous aprotic KPF₆-CH₃CN solution (*J. Am. Chem. Soc.*, 2018, 140, 2397–2400), which are compared with those of Pt/NF.

The N₂ adsorption-desorption isotherms of Ni₃N/Ni/NF and Pt/NF (loading mass of Pt/C: 1.5 and 2.5 mg cm⁻² on Ni foam) were measured (see Supplementary Information for details) and shown in **Figure R11**. Their specific BET surface areas are summarized in **Table R1**. It should be noted that we optimized the mass loading of Pt/C on Ni foam and found that 2.5 mg cm⁻² of Pt/C showed the best HER performance while 1.5 mg cm⁻² exhibited the highest HOR activity (see **Figures R20-R21** for detailed comparison).

Table R1. The specific BET surface areas of Ni₃N/Ni/NF and Pt/NF.

Sample	Specific BET surface area (m ² g ⁻¹) ^a
Ni ₃ N/Ni/NF	1.61
Pt/NF (1.5 mg cm ⁻²)	6.22
Pt/NF (2.5 mg cm ⁻²)	4.47

^aThe specific BET area is normalized by the total mass of Ni₃N/Ni/NF or Pt/NF.

Figure R11. N₂ adsorption-desorption isotherms of Ni₃N/Ni/NF, Pt/NF (1.5 mg cm⁻²) and Pt/NF (2.5 mg cm⁻²).

Then the current densities of Ni₃N/Ni/NF and Pt/NF for HER were normalized by their BET surface areas for a fair comparison of their intrinsic specific activities. The polarization curves of Ni₃N/Ni/NF and Pt/NF for HER in 1.0 M KOH and 1.0 M KPi normalized by their BET specific surface areas are shown in **Figure R12**, respectively. Apparently, Ni₃N/Ni/NF exhibited much higher intrinsic specific activity than Pt/NF for HER in both alkaline and neutral solutions. The normalized polarization curves of

Ni₃N/Ni/NF and Pt/NF for HOR in H₂-saturated 0.1 M KOH are also compared (**Figure R13**), also confirming the higher intrinsic specific activity of Ni₃N/Ni/NF.

Figure R12. The BET surface area-normalized linear sweep voltammetry curves of Pt/NF (2.5 mg cm⁻²) and Ni₃N/Ni/NF for HER in 1.0 M KOH (a) and 1.0 M KPi (b).

Figure R13. The BET surface area-normalized steady-state polarization curves of Pt/NF (1.5 mg cm⁻²) and Ni₃N/Ni/NF for HOR in H₂-saturated 0.1 M KOH.

The electrochemically active surface areas (ECSAs) of Ni₃N/Ni/NF and Pt/NF were also estimated via the double layer capacitance method by collecting their cyclic voltammetry (CV) curves at different scan rates in aprotic CH₃CN with 0.15 M KPF₆ (**Figure R14**, see Supplementary Information for details), which minimizes the ion transfer reaction to provide more accurate ECSA values for comparison (*J. Am. Chem. Soc.*, 2018, 140, 2397–2400). The obtained ECSA values are summarized in **Table R2**.

Figure R14. CV curves of (a) Ni₃N/Ni/NF, (b) Pt/NF (2.5 mg/cm²), and (c) Pt/NF (1.5 mg/cm²) collected at various scan rates ranging from 4 to 20 mV s⁻¹ and (d) their linear fittings in CH₃CN with 0.15 M KPF₆.

Table R2. The geometric areas and electrochemically active surface areas (ECSAs) of Ni₃N/Ni/NF and Pt/NF.

Sample	Geometric area (cm ²)	ECSA (cm ²)
Ni ₃ N/Ni/NF	0.25	247
Pt/NF (1.5 mg cm ⁻²)	0.25	450
Pt/NF (2.5 mg cm ⁻²)	0.25	680

Then the current densities of Ni₃N/Ni/NF and Pt/NF for HER and HOR were normalized by their corresponding ECSAs, resulting a fair comparison of their specific activities. As shown in **Figures R15-R16**, Ni₃N/Ni/NF still showed higher catalytic current densities than Pt/NF for both HER and HOR.

Therefore, both the apparent activities (normalized by the geometric electrode area) and intrinsic specific activities (normalized by either BET surface area or ECSA) of Ni₃N/Ni/NF for HER in neutral and alkaline solutions and HOR in alkaline solution are indeed higher than those of Pt/NF under the same conditions.

Figure R15. The ECSA-normalized linear sweep voltammetry curves of optimized Pt/NF (2.5 mg/cm²) and Ni₃N/Ni/NF for HER in 1.0 M KOH (a) and 1.0 M KPi (b).

Figure R16. The ECSA-normalized steady-state polarization curves of optimized Pt/NF (1.5 mg/cm²) and Ni₃N/Ni/NF for HOR in H₂-saturated 0.1 M KOH.

(5) In Fig. 2a and 2c, the current densities is positive when the potential is zero vs. RHE. Please explain it. Please offer the HER performance of Ni₃N/NF.

Our response: We have removed the HER study of Ni₃N/Ni/NF in the acidic electrolyte because of its limited long-term stability. In 1.0 M KOH, the current density of Ni₃N/Ni/NF was slightly positive at 0 V vs. RHE, possibly due to adventitious capacitance background current in the polarization curve scanned from negative to positive potential, which has been found previously (*Nature Mater.*, 2012, 11, 963–969; *Energy Environ. Sci.*, 2018, 11, 744–771; *Angew. Chem. Int. Ed.*, 2014, 53, 14433–14437; *J. Am. Chem. Soc.*, 2013, 135, 9267–9270). The HER activity of Ni₃N/Ni/NF in 1.0 M KOH has been measured again and the current density of Ni₃N/Ni/NF in 1.0 M KOH at 0 V vs. RHE is nearly 0 mA cm⁻² (Figure 2b in the main text).

Following the reviewer’s suggestion, a pristine Ni foam (NF) was also directly nitridized by NH₃ to obtain Ni₃N/NF under the same conditions as Ni₃N/Ni/NF. The SEM image of Ni₃N/NF (Figure R4) shows smooth surface with uniform nickel and nitrogen distribution. The HER polarization curves of Ni₃N/NF were measured under the same conditions and compared with those of Ni₃N/Ni/NF (Figure R6). The electrocatalytic HER activities of Ni₃N/NF in alkaline and neutral solutions are much lower than those of Ni₃N/Ni/NF.

Referee # 2:

This manuscript reports a plausible Ni₃N/Ni electrocatalyst with certain HER/HOR performance in aqueous electrolytes at different pH values, in combination with density functional theory computations. The claims are pretty interesting and if the claims were correct and convincing, they would be nice contributions to a wide community. My concerns are listed in the following:

Our response: We appreciate the reviewer’s comments that help to improve the quality of our manuscript. More control experiment results have been collected to support our conclusions.

I am not sure if the interface of Ni₃N/Ni does exist and is available for HER/HOR merely based on SEM and TEM images. Detailed structural characterizations, such as high-resolution (sub-nanometer) elemental mapping, are required to investigate the formation of Ni₃N/Ni and to quantify the coverage

percentage of Ni₃N on the Ni substrate. In addition, is it possible to experimentally determine the thickness of Ni₃N?

Our response: High resolution TEM (HRTEM) image and sub-nanometer elemental mapping results are shown in **Figures R1b** and **R2**, respectively. The lattice fringes were carefully examined with fast Fourier transform (FFT). The HRTEM image clearly shows the interface between hexagonal Ni₃N and cubic Ni. The well-resolved lattice fringes with inter-planar spacing of 0.204 and 0.214 nm can be unambiguously assigned to the (111) and (002) crystal planes of hexagonal Ni₃N (JCPDS Card No. 10-0280) with an intersection angle of 62°. The unique lattice fringes with inter-planar distance of 0.176 nm correspond to the (200) crystal plane of cubic Ni (JCPDS Card No. 04-0850). Moreover, the elemental maps of Ni and N exhibit that Ni is homogeneously distributed, while N is sporadically located on the surface of Ni₃N/Ni/NF. Overall, our new HRTEM and elemental mapping results corroborate the formation of Ni₃N/Ni interface.

Figure R17. The weight percent of Ni₃N in Ni₃N/Ni/NF subjected to NH₃ treatment at 300 °C for different durations.

It is challenging to precisely measure the thickness of Ni₃N, as the surface of Ni₃N/Ni is very rough. Instead, the weight percentage of Ni₃N in Ni₃N/Ni/NF synthesized with different nitridation durations could be obtained on the basis of the weight increment after NH₃ treatment. The weight percentage of Ni₃N in Ni₃N/Ni/NF increased from 8.67 to 44.66 wt.%, as the nitridation time at 300 °C rose from 0.5 to 12 h, indicating the increased coverage of Ni₃N (**Figure R21**). Among these samples, the Ni₃N/Ni/NF synthesized at 300 °C for 6 h with the weight percentage of Ni₃N of 41.82 wt.% showed the best HER activity in neutral and alkaline electrolytes (**Supplementary Fig. 6-7**), highlighting the importance in obtaining the appropriate amount of Ni₃N/Ni interfacial sites for optimal HER performance.

Ag/AgCl was employed as the reference electrode in this study. However, the authors did not specify if a salt bridge was used to separate Ag/AgCl from the electrolyte. If not, chloride would seriously poison the surface of platinum, thus underestimating the electrochemical performance of platinum and losing a fair comparison. In this case, all of the claims about electrochemical performance will not make sense.

Our response: We agree with the reviewer that the electrochemical performance of platinum should be reliable for fair comparison. A salt bridge kit (Pine Research) was used to separate the Ag/AgCl (Pine Research) reference electrode from the electrolytes, which was calibrated prior to each experiment. In order to avoid any potential poisoning of Pt catalyst from chloride, Hg/HgO (1.0 M KOH, CH Instruments) and Hg/Hg₂SO₄ (saturated K₂SO₄, CH Instruments) reference electrodes were also used to verify the electrocatalytic performance of Pt/NF in 1.0 M KOH and KPi solutions, respectively. There is

no discrepancy among the results of Pt/NF measured by either Hg/HgO or Hg/Hg₂SO₄ and the previous results measured using Ag/AgCl coupled with the salt bridge kit for HER and HOR (**Figures R18-19**). It is worth noting that the HER activity of our optimal Pt/NF presented in this work (*e.g.*, overpotential required to deliver a benchmark current density of -10 mA cm^{-2} , $\eta_{10} = 15.3 \text{ mV}$ in 1.0 M KOH) stands out among state-of-the-art Pt catalysts under similar conditions in the literature, which show a wide variation in HER activities (*Nat. Commun.*, 2015, 6, 6430; *Nat. Nanotechnol.*, 2017, 12, 441-446; *Nat. Commun.*, 2016, 7, 13638; *Chem*, 2017, 3, 122-133; *J. Am. Chem. Soc.*, 2016, 138, 16174-16181; *Nat. Commun.*, 2017, 8, 14969; *Energy Environ. Sci.*, 2018, DOI: 10.1039/C7EE03603E). Therefore, the high catalytic HER performance of Ni₃N/Ni/NF compared to Pt/NF in neutral and alkaline solutions is intrinsic, and not because a poor Pt/NF benchmark was used.

Figure R18. Linear sweep voltammetry curves of Pt/NF (2.5 mg cm^{-2}) for HER in 1.0 M KOH (a) and 1.0 M KPi (b) using Ag/AgCl or Hg/HgO as the reference electrode

Figure R19. Steady-state polarization curves Pt/NF (1.5 mg cm^{-2}) for HOR in H₂-saturated 0.1 M KOH using Ag/AgCl or Hg/HgO as the reference electrode.

It is critical to use the best performance of platinum as the standard, so the data on the optimization of Pt/C toward HER/HOR on Ni foam are required.

Our response: We agree that it is critical to use the best performance of Pt as the standard. The optimization results of Pt/NF with different loadings of Pt/C on NF towards HER and HOR are shown in **Figures R20-21**. The optimal loading of Pt/C on NF for HER was 2.5 mg cm^{-2} , while for HOR was 1.5 mg cm^{-2} .

Figure R20. Linear sweep voltammetry curves of Pt/NF with different mass loadings measured for HER in 1.0 M KOH (a) and 1.0 M KPi (b).

Figure R21. Steady-state polarization curves Pt/NF with different mass loadings measured for HOR in H_2 -saturated 0.1 M KOH.

I cannot tell the HOR part is a good addition or else, since no DFT study was carried out for HOR.

Our response: Great efforts have been devoted to the development of nonprecious HER electrocatalysts and significant progress has been achieved in the transition metal based HER catalysts. **However, much less attention has been paid to the investigation of nonprecious electrocatalysts for alkaline HOR, which are of critical importance to the development of hydroxide exchange membrane fuel cells. Therefore, we believe that the HOR study of $\text{Ni}_3\text{N}/\text{Ni}/\text{NF}$ is a good addition, inspiring more research in this key area.** Furthermore, it has been rarely reported for Earth-abundant electrocatalysts like $\text{Ni}_3\text{N}/\text{Ni}/\text{NF}$ showing **bifunctional activities for both HER and HOR.**

It has been proposed that the hydrogen (H^*) adsorption free energy (ΔG_{H^*}) is regarded as a key descriptor in determining HOR activity (*Nat. Commun.*, 2015, 6, 5848; *Nat. Commun.*, 2016, 7, 10141; *Energy Environ. Sci.*, 2014, 7, 1719-1724; *Sci. Adv.*, 2016, 2, e1501602). The best hydrogen adsorption site on $Ni_3N/Ni/NF$ exhibits a ΔG_{H^*} value of 0.01 eV, very close to the ideal value of 0 eV, which implies its high activity for HOR as well. The adsorption energies of H_2 on Ni_3N/Ni interface, Ni_3N , and Ni were also calculated by DFT computations. Ni_3N/Ni interface and Ni show similar H_2 adsorption energies, which are significantly higher than that on Ni_3N (0.05 – 0.06 eV), indicating the favored H_2 adsorption on Ni_3N/Ni interfacial sites and Ni (**Supplementary Fig. 54-60**). Given the ideal H adsorption energy and the strong H_2 adsorption on Ni_3N/Ni , it is not surprising that $Ni_3N/Ni/NF$ exhibits superior HOR performance. Although fully taking the solvent environment into account when calculating the adsorption of H^* and H_2 is very challenging and beyond the scope of this work, our ongoing work aims to compute the apparent H adsorption energy, which has been recently proposed as a pH-dependent descriptor for HER and HOR activities (*J. Electrochem. Soc.*, 2018, 165, H27-H29).

Normally, the performance of Pt/C becomes lower and lower with the increase of pH values, why Pt/C in this study did not follow such a trend?

Our response: The HER performance of Pt/C will decrease along the increase of pH if the electrolytes have the same conductivity. However, in our case, the conductivity of 1.0 M KPi (95.3 mS/cm) is substantially lower than that of 1.0 M KOH (217 mS/cm). Therefore, the measured HRE performance of Pt/C in 1.0 M KPi was lower than that in 1.0 M KOH. The same trend has been reported in many publications (*Adv. Mater.*, 2017, 29, 1606521; *Angew. Chem. Int. Ed.*, 2017, 56, 11559–11564; *Adv. Mater.*, 2017, 29, 1605502; *Chem. Sci.*, 2018, 9, 1970–1975; *ACS Sustainable Chem. Eng.*, 2018, 6, 6388–6394).

The same references are repetitively listed and should be corrected.

Our response: The repetitive references have been removed and all of the references are carefully examined.

Is the labeling of supplemental Figure 14 correct?

Our response: The labelling of supplemental Figure 14 has been corrected.

For HER plots, if iR-correction has been done?

Our response: All LSV polarization curves for HER were iR-corrected, unless stated otherwise. The correction was made according to the following equation (*Chem. Mater.*, 2017, 29, 120–140):

$$E_{\text{corrected}} = E_{\text{measured}} - iR_s$$

where $E_{\text{corrected}}$ is the iR-corrected potential, E_{measured} and i are experimentally measured potential and current, respectively, and R_s is the equivalent series resistance measured by electrochemical impedance spectroscopy (EIS) in the frequency range of $10^6 - 0.1$ Hz with an amplitude of 10 mV.

Referee # 3:

In this work the authors report on the fabrication of Ni_3N/Ni foam electrode as electrocatalyst for hydrogen evolution and oxidation reaction. The material and the synthesis are not new but the performance and mechanistic insights are very interesting. Therefore, I recommend its publication after addressing the following points:

1. Many works show the direct growth of Ni_3N on nickel foam as electrocatalyst for HER/OER. In these works, also they can obtain the Ni/Ni_3N structure. The authors should compare their results to a direct

Ni₃N growth. Moreover, as they create high surface area network Ni₃N particles - a comparison of the Ni₃N on carbon foam is required to prove the synergetic effect with Ni foam.

Our response: Supplementary Table 3 compares our results with those of various reported Ni₃N-based HER electrocatalysts in situ grown or coated on different current collectors including Ni foam, Ni mesh, Ti mesh, Ti foil and glassy carbon electrode, which unambiguously demonstrates that the catalytic HER performance of Ni₃N/Ni/NF is superior to those of all the reported Ni₃N counterparts.

We also directly utilized a pristine Ni foam (NF) for thermal NH₃ treatment to produce Ni₃N/NF. The SEM image of Ni₃N/NF (**Figure R4**) shows smooth surface, consistent with the previous report (*Electrochim. Acta*, 2016, 191, 841–845). In contrast, Ni₃N/Ni/NF possesses macroporous ligament network structure with numerous stacked coarse particles over the skeleton surface (**Figure 1a** in the main text). The comparison of the nitrogen adsorption-desorption isotherms (**Figure R5**) shows that Ni₃N/Ni/NF has a larger BET surface area than Ni₃N/NF. The electrocatalytic HER activities in alkaline and neutral solutions of Ni₃N/NF as well as its HOR performance in 0.1 M KOH are inferior to those of Ni₃N/Ni/NF (**Figure R6**), indicating the advantages of using Ni/NF (electrodeposited Ni microspheres coated Ni foam) instead of a pristine Ni foam as the Ni source.

Following the reviewer's suggestions, we also fabricated Ni₃N/Ni on carbon foam (CF) to obtain Ni₃N/Ni/CF with the same synthetic approach except replacing Ni foam with carbon foam. The Ni₃N/Ni/CF was characterized by XRD (**Figure R1a**), SEM (**Figure R7**), and XPS (**Figure R8**), confirming the presence of Ni₃N/Ni over CF with similar composition and morphology as those of Ni₃N/Ni/NF.

The HER performance of Ni₃N/Ni/CF was measured in 1.0 M KOH and 1.0 M KPi and compared with that of Ni₃N/Ni/NF. The linear sweep voltammetry and chronopotentiometry curves of Ni₃N/Ni/CF almost overlap with those of Ni₃N/Ni/NF under the same conditions (**Figures R9**), demonstrating that the Ni₃N/Ni interfacial sites on CF have similar catalytic activity and stability as those on NF.

The HOR performance of Ni₃N/Ni/CF was also measured under the same condition as for Ni₃N/Ni/NF. In contrast to the negligible current obtained in the Ar-saturated 0.1 M KOH, Ni₃N/Ni/CF shows appreciable anodic current beyond 0 V vs. RHE upon H₂ saturation (**Figure R10a**), suggesting that Ni₃N/Ni/CF also has high catalytic activity towards HOR. The long-term stability of Ni₃N/Ni/CF for HOR was assessed by chronoamperometry at 0.09 V vs. RHE in H₂-saturated 0.1 M KOH (**Figure R10b**). The initial HOR current density of Ni₃N/Ni/CF was 3.64 mA cm⁻² and gradually decreased to 2.72 mA cm⁻² after 24 h. Note that the HOR current density of Ni₃N/Ni/CF is lower than that of Ni₃N/Ni/NF under the same condition. It can be rationalized that NF serves as a favorable H₂ gas diffusion layer besides the roles of current collector and catalyst support. HOR occurs at a three-phase interface (gaseous H₂–liquid KOH electrolyte–solid Ni₃N/Ni catalyst), in which H₂ diffusion plays an important role in the measured current (*Nat. Commun.*, 2014, 6, 5848; *Sci. China Mater.*, 2016, 59, 217–238; *Adv. Mater.*, 2017, 29, 1604685). Therefore, the gas diffusion layer greatly influences the measured HOR current. The metallic Ni foam has a 3D macroporous network structure with better porosity and mechanical robustness than that of carbon foam composed of interconnected carbon fibers, allowing Ni₃N/Ni/NF (**Figure 1** in the main text) to possess beneficial porosity for H₂ diffusion (**Figure R7**). It was reported that the metal-based gas diffusion layers have advantages over carbon-based diffusion layers in gas electrodes (*Adv. Mater.*, 2017, 29, 1604685). As a result, the robust Ni foam may facilitate H₂ diffusion, accounting for the higher current density and better stability of Ni₃N/Ni/NF for HOR than those of Ni₃N/Ni/CF.

In summary, Ni foam serves as a robust current collector and support substrate for Ni₃N/Ni interfacial catalytic sites for electrocatalytic HER and HOR and acts as a macroporous gas diffusion medium for facilitating H₂ gas diffusion in HOR. Similar HER activity could be obtained for Ni₃N/Ni/CF, albeit it shows lower HOR performance.

2. Can they show if they control the Ni₃N thickness/amount and how does it affect the catalytic properties?

Our response: The amount of Ni₃N in Ni₃N/Ni/NF can be controlled by the thermal nitridation duration. The weight percentage of Ni₃N in Ni₃N/Ni/NF synthesized with different nitridation durations could be obtained on the basis of the weight increase after NH₃ treatment. The weight percentage of Ni₃N in Ni₃N/Ni/NF increased from 8.67 to 44.66 wt.%, as the nitridation time at 300 °C rose from 0.5 to 12 h, indicating the increased coverage of Ni₃N (**Figure R17**). Among these samples, the Ni₃N/Ni/NF synthesized at 300 °C for 6 h with the weight percentage of Ni₃N as 41.82 wt.% possessed the best HER activity in all electrolytes (**Supplementary Fig. 6-7**), highlighting the importance in obtaining the appropriate amount of Ni₃N/Ni interfacial sites for optimal HER performance.

3. The authors should provide some post-characterizations as XPS, XRD, etc. to further support the high stability (especially in acid).

Our response: We found that Ni₃N/Ni/NF showed excellent HER activity in 0.5 M H₂SO₄ during the short duration of LSV measurement, even though the underlying Ni component might gradually dissolve in acidic solution. Actually, Ni foams coated with various in situ grown transition metal based catalysts such as Ni_{1-x}Co_xSe₂ (*Adv. Mater.*, 2017, 29, 1606521), Ni₃S₂ (*J. Am. Chem. Soc.*, 2015, 137, 14023–14026), Co-Ni-P (*J. Mater. Chem. A*, 2016, 4, 10195-10202; *Catal. Today*, 2017, 287, 122-129), and Ni₂P (*ACS Appl. Mater. Interfaces*, 2015, 7, 2376–2384) have been reported to show HER activities in 0.5 M H₂SO₄ within the limited electrochemical characterization duration. It is speculated that the coating layer with high HER activity and relatively good resistivity against acidic corrosion will slow down (if not completely prohibit) the dissolution of Ni foam. However, considering the limited long-term stability of Ni foam or catalyst-coated Ni foam in acidic media, the catalytic HER study of Ni₃N/Ni/NF in the acidic electrolyte has been removed and we focus on emphasizing the unprecedented HER activity and stability of Ni₃N/Ni/NF in the neutral and alkaline electrolytes and the activity and stability for HOR in 0.1 M KOH.

The XRD patterns and XPS spectra of Ni₃N/Ni/NF after extended HER or HOR electrolysis in neutral or alkaline electrolyte were collected. **Figure R22** presents the XRD patterns of Ni₃N/Ni/NF, which retained the original crystallinity after long-term electrolysis. The XPS spectra of the post-catalysis Ni₃N/Ni/NF also exhibit little changes compared to that of pristine Ni₃N/Ni/NF (**Figure R23**). These post-electrolysis characterization results corroborate the strong robustness of Ni₃N/Ni/NF for HER and HOR.

Figure R22. XRD patterns of Ni₃N/Ni/NF after catalyzing HER in 1.0 M KOH or KPi solution for 50 h and HOR in 0.1 M KOH for 24 h.

Figure 23. XPS spectra of Ni 2p_{3/2} and N 1s of Ni₃N/Ni/NF before and after catalyzing HER or HOR.

4. The authors should further elaborate the Tafel slope values in KOH. What is the mechanism for the Ni₃N/Ni?

Our response: Tafel analysis was conducted on the HER polarization curves of Pt/NF, Ni₃N/Ni/NF, and Ni/NF in 1.0 M KOH. As shown in the Tafel plots (**Figure R24**), the logarithm of HER current density does not increase linearly with the overpotential (η) in the entire measured potential window. The Tafel slopes of Pt/NF and Ni₃N/Ni/NF would change if the selected overpotential (η) range is varied for linear fitting. For instance, Ni₃N/Ni/NF showed a small Tafel slope of 29.3 mV/dec within the range of $\eta = 9 - 18$ mV; while a larger fitted slope of 80.1 mV/dec was obtained in a higher overpotential region of 49 to 74 mV. A similar trend of the potential-dependent Tafel slope of Pt/NF was also observed, which shifted from 32.2 ($\eta = 9 - 24$ mV) to 117 mV/dec ($\eta = 46 - 80$ mV).

The findings of the Tafel plots with variable slopes were reported in many HER catalysts including Co_{0.6}Mo_{1.4}N₂ (*J. Am. Chem. Soc.*, 2013, 135, 19186–19192), Pt (*Angew. Chem. Int. Ed.*, 2012, 51, 12703–12706; *J. Electrochem. Soc.*, 1952, 99, 488-494; *J. Electrochem. Soc.*, 1954, 101, 426-432; *J. Chem. Soc., Faraday Trans.*, 1996, 92, 3719-3725), Pt-rare earth alloy (*Int. J. Hydrogen Energy*, 2013, 38, 3137-3145), Ni (*J. Chem. Phys.*, 1952, 20, 614; *Int. J. Hydrogen Energy*, 1995, 20, 435-440; *J. Am. Chem. Soc.*, 2017, 139, 4854–4858), Ni-Mo (*J. New Mat. Electrochem. Syst.*, 2010, 13, 239-244), Ni₂P (*J. Am. Chem. Soc.*, 2013, 135, 9267–9270; *Phys. Chem. Chem. Phys.*, 2015, 17, 10823–10829), MoP (*Angew. Chem. Int. Ed.*, 2014, 53, 14433–14437) and FeP/Ni₂P (*Nat. Commun.*, 2018, 9, 2551) under different pH conditions. The variable Tafel slopes could be attributed to several factors, such as back reaction at low overpotentials, mass transport together with the blocking effect of produced H₂ bubbles at high overpotentials, formation of a large number of N-H moieties, and the dependence of adsorbed H^{*} coverage on overpotential (*J. Am. Chem. Soc.*, 2013, 135, 19186–19192; *Angew. Chem. Int. Ed.*, 2012, 51, 12703–12706). It has been reported that Tafel slope is potential dependent and in turn H^{*} coverage dependent (*Sci. Rep.*, 2015, 5, 13801; *Int. J. Hydrogen Energy*, 1995, 20, 435-440; *J. Electroanal. Chem.*, 1986, 198, 149-175). Furthermore, an electrocatalyst of considerable surface roughness may also lead to large deviations from an ideal model for theoretical Tafel slope determination.

Bockris *et al.* has identified the Tafel plots similar to that of Ni₃N/Ni/NF in this work on Ni, Ag and Pt cathodes for HER (*J. Electrochem. Soc.*, 1952, 99, 169-186; *J. Chem. Phys.*, 1952, 20, 614). Conway and coworkers have systematically investigated the variation of Tafel slopes dependent on the overpotential over the HER cathodes of Ni, Ni-Mo, Ni-Mo-Cd and Pt and characterized the coverage and pseudocapacitance behavior of overpotential-deposited (OPD) H species in the cathodic HER process (*J. Electrochem. Soc.*, 1983, 130, 1825-1836; *J. Electroanal. Chem.*, 1984, 161, 39-49; *J. Chem. Soc.*

Faraday Trans., 1985, 81, 1841-1862; *Int. J. Hydrogen Energy*, 1986, 11, 533-540; *Electrochim. Acta*, 1986, 31, 1013-1024; *J. Electroanal. Chem.*, 1986, 198, 149-175; *Int. J. Hydrogen Energy*, 1987, 12, 607-621). They pointed out that the change of the Tafel slope with the increase of overpotential indicates a change of mechanism between two consecutive elementary steps in a reaction pathway of HER (e.g., Volmer step followed by the parallel Tafel and Heyrovsky steps) and/or a change of adsorption behavior of H^* with potential (i.e., potential-dependent H^* coverage and H^* pseudocapacitance) as well as the formation of surface-phase hydride.

Therefore, although Tafel analysis has been used to probe the mechanism of HER for certain electrocatalysts (mostly in specifically selected overpotential regions), it is difficult to apply such analysis in the present work due to the variable Tafel slope values dependent on overpotential, as indicated in other novel HER catalysts (*Angew. Chem. Int. Ed.*, 2012, 51, 12703–12706; *J. Am. Chem. Soc.*, 2013, 135, 19186–19192). A rigorous mechanistic microkinetic analysis will be conducted on crystalline thin films for Ni_3N/Ni (*J. Electrochem. Soc.*, 2015, 162, F1470-F1481; *J. Am. Chem. Soc.*, 2017, 139, 4854–4858), which will be the subject of our future research.

Figure R24. The Tafel plots of (a) $Ni_3N/Ni/NF$, (b) optimized Pt/NF and (c) Ni/NF derived from their respective polarization curves measured in 1.0 M KOH.

5. The theoretical part is very interesting - how is the mechanism compared to Pt electrode? In addition, it is still not really clear why a Tafel mechanism is observed in 1M KOH. Due to the nitrogen sites?

Our response: Figure R24 illustrates the Tafel plots of $Ni_3N/Ni/NF$, Pt/NF and Ni/NF derived from their respective polarization curves collected in 1.0 M KOH. The Tafel slopes of both $Ni_3N/Ni/NF$ and Pt/NF increase monotonically with the overpotential. The potential-dependent Tafel slopes make it difficult to precisely elucidate the kinetic mechanism and identify the rate-limiting steps of $Ni_3N/Ni/NF$ and Pt/NF for HER. However, the Tafel slope of $Ni_3N/Ni/NF$ is lower than that of Pt/NF , when they are compared in the same overpotential region. In the alkaline solution, the Volmer step is generally regarded as the initial step ($H_2O + e^- + * \leftrightarrow H^* + OH^-$, where $*$ denotes a free active site), involving the water dissociation process (*J. Am. Chem. Soc.*, 2017, 139, 4854–4858). The Tafel analysis (Figure R24) indicates that the Ni_3N/Ni interface favors the dissociative adsorption of water compared to Pt and Ni under the present experimental conditions. Nevertheless, at the current stage, it is difficult to ascertain the

rate determining step(s) and intrinsic kinetic mechanism of Ni₃N/Ni/NF for HER on the basis of its potential-dependent Tafel slopes. Future work will aim to elucidate the intrinsic kinetic mechanism of the Ni₃N/Ni interfacial catalysts.

REVIEWERS' COMMENTS:

Reviewer #1 (Remarks to the Author):

Although the authors have addressed most concerns raised by other reviewers, I still think this work suffers from limited novelty.

Reviewer #2 (Remarks to the Author):

The authors have well addressed all the concerns. Now the manuscript is publishable. However, the electrochemical performance on Ni foam is likely different from that on RDE or other substrates. The authors should carefully draw the conclusion that their material is superior to commercial Pt or Pt/C.

Reviewer #3 (Remarks to the Author):

I have carefully read the reviewer comments and the authors responses. I think that the current manuscript is significantly improved and that it is now suitable for publication.

Point-by-Point Response Letter

Referee # 1:

Although the authors have addressed most concerns raised by other reviewers, I still think this work suffers from limited novelty.

Our response: We would like to emphasize the novelty and significance of our work beyond the state of the art to justify its publication.

Although many Ni₃N materials and composites have been developed through sophisticated synthetic methods and used as electrocatalysts for HER, their electrocatalytic performances in those reports are still much inferior to those of Pt catalysts (Supplementary Table 3). To the best of our knowledge, few catalytic systems reported so far have focused on exploring the interfaces of 1st-row transition metals and their nitrides for hydrogen electrochemistry (both hydrogen evolution and oxidation reactions) in aqueous media, even though some Ni₃N-based composites have been reported for only catalyzing HER in either neutral or alkaline solution, such as Ni₃N/carbonaceous substrates (*J. Mater. Chem. A*, 2017, 5, 9377–9390; *J. Mater. Chem. A*, 2016, 4, 17363–17369; *Inorg. Chem. Front.*, 2017, 4, 1120–1124; *J. Mater. Chem. A*, 2015, 3, 8171–8177), Ni₃N/nickel borate (*J. Mater. Chem. A*, 2017, 5, 7806–7810), Ni₃N/TiN (*J. Mater. Chem. A*, 2016, 4, 5713–5718), Ni₃N/NiMoN (*Nano Energy*, 2018, 44, 353–363) and Ni₃N/Pt (*Adv. Energy Mater.*, 2017, 7, 1601390). HER is indeed important for sustainable H₂ production, however, HOR is of vital importance for H₂ utilization in hydrogen fuel cells that produce electricity with water as the sole product. The success of a future “hydrogen economy” strongly relies on both of the efficient H₂ production and utilization. In this context, **efficient hydrogen production and utilization materials will be crucial in order to compete with fossil fuel technologies.**

However, the catalytic activity of Ni₃N towards HOR still remains unexplored. None of the above publications comprehensively investigate the effects of transition metal/transition metal nitride interfaces on some critical properties governing the H₂ electrochemistry, including hydrogen adsorption free energy (ΔG_{H^*}), adsorption energy of water, energy barrier of water dissociation, and adsorption energy of H₂.

Our work demonstrates that rationally interfacing Ni and Ni₃N results in a competent electrocatalyst (Ni₃N/Ni) with extraordinary activities for both HER in neutral and alkaline electrolytes and HOR in alkaline solution. The novelty and significance of our work are shown as follows.

(i) The abundant Ni₃N/Ni interfacial sites are fabricated by facile electrodeposition of porous Ni particles on common current collectors of Ni foam or carbon foam, followed by controllable thermal nitridation in NH₃. The simple cathodic electrodeposition method endows the electrocatalyst with hierarchical topology and highly porous architecture, while the amount of Ni₃N can be readily tuned by adjusting thermal nitridation time. The electrodeposition and nitridation operations can be carried out under atmospheric pressure in a simple and scalable manner which is favorable for large-scale industrial production, whereas fabrication of reported Ni₃N composite catalysts usually involves sophisticated high-pressure hydrothermal/solvothermal procedures (see references above). Moreover, the roles of rough Ni/NF structure and Ni foam substrate as well as the effects of Ni₃N amount in Ni₃N/Ni/NF on the electrocatalytic activities for HER or HOR have been explicitly investigated in our current work.

(ii) The rich Ni₃N/Ni interfacial sites offer extraordinary electrocatalytic activities for both HER and HOR, which can rival those of Pt electrocatalysts loaded on NF in terms of both apparent activities (normalized by the geometric electrode areas) and intrinsic specific activities (normalized by both BET and electrochemically active surface areas) together with remarkable durability under similar experimental measurement conditions within the scope of our investigation. The performances of such a Ni₃N/Ni

electrocatalyst towards HER in neutral and alkaline solutions are substantially better than those of reported Ni₃N and Ni₃N-based composite electrocatalysts (Supplementary Table 3).

(iii) Moreover, the HOR performance of our Ni₃N/Ni electrocatalyst in 0.1 M KOH is also remarkable, demonstrating zero overpotential of catalytic onset, robust long-term stability, and outstanding CO tolerance, which has never been reported for nonprecious electrocatalysts to the best of our knowledge. Particularly, high tolerance to CO impurity is a desirable property of electrocatalysts in hydrogen fuel cells, as the current cost-effective industrial production of H₂ is mainly relied on steam reforming from hydrocarbons, which may result in CO impurity in the final H₂ gas.

(iv) Finally, our DFT computation results elucidate the effects of Ni₃N/Ni interface on some crucial properties regulating the hydrogen electrochemistry, including ΔG_{H^*} , adsorption energy of H₂O, energy barrier of water dissociation and adsorption energy of H₂, which will provide guidance for the rational design of inexpensive and highly efficient HER electrocatalysts and advance the development of nonprecious HOR electrocatalysts.

Therefore, we believe that the novelty and significance of this work meet the high standards of *Nature Communications* very well. We sincerely hope the reviewer would kindly consider the manuscript that reflects the aforementioned novelty and significance.

Referee # 2:

The authors have well addressed all the concerns. Now the manuscript is publishable. However, the electrochemical performance on Ni foam is likely different from that on RDE or other substrates. The authors should carefully draw the conclusion that their material is superior to commercial Pt or Pt/C.

Our response: We thank the reviewer for the comments. We agree that the electrochemical performance of Pt on Ni foam could be different from that on rotating disk electrodes. We have revised the description to cautiously draw the conclusion that the electrocatalytic performance of our interfacial Ni₃N/Ni electrocatalysts on Ni foam for HER and HOR can rival the activities of Pt/C catalysts loaded on Ni foam under similar experimental conditions within the scope of our investigation.

Referee # 3:

I have carefully read the reviewer comments and the authors responses. I think that the current manuscript is significantly improved and that it is now suitable for publication.

Our response: We greatly appreciate the reviewer's comments and valuable suggestions.